# Tp63-expressing adult epithelial stem cells cross lineages boundaries revealing latent hairy skin competence

Stéphanie Claudinot [1,2✉], Jun-Ichi Sakabe[3,4,5], Hideo Oshima[6], Christèle Gonneau[1,2], Thimios Mitsiadis [7], Daniel Littman[1,2], Paola Bonfanti[8], Geert Martens[9], Michael Nicolas [1,2], Ariane Rochat[1,2] & Yann Barrandon [1,2,3,4,5✉]

The formation of hair follicles, a landmark of mammals, requires complex mesenchymal–epithelial interactions and it is commonly believed that embryonic epidermal cells are the only cells that can respond to hair follicle morphogenetic signals in vivo. Here, we demonstrate that epithelial stem cells of non-skin origin (e.g. that of cornea, oesophagus, vagina, bladder, prostate) that express the transcription factor Tp63, a master gene for the development of epidermis and its appendages, can respond to skin morphogenetic signals. When exposed to a newborn skin microenvironment, these cells express hair-follicle lineage markers and contribute to hair follicles, sebaceous glands and/or epidermis renewal. Our results demonstrate that lineage restriction is not immutable and support the notion that all Tp63-expressing epithelial stem cells, independently of their embryonic origin, have latent skin competence explaining why aberrant hair follicles or sebaceous glands are sometimes observed in non-skin tissues (e.g. in cornea, vagina or thymus).

[1] Laboratory of Stem Cell Dynamics, School of Life Sciences, Ecole Polytechnique Fédérale Lausanne, 1015 Lausanne, Switzerland. [2] Centre Hospitalier Universitaire Vaudois, 1011 Lausanne, Switzerland. [3] Duke-NUS Graduate Medical School, Singapore, Singapore. [4] Department of Plastic, Reconstructive and Aesthetic Surgery, Singapore General Hospital, Singapore, Singapore. [5] Skin Research Institute of Singapore A*STAR, Singapore, Singapore. [6] Department of Plastic Surgery, National Hospital Organization, Kumamoto Medical Centre, 1-5 Ninomaru Kumamoto, 860-0008 Kumamoto, Japan. [7] Institute of Oral Biology, University of Zurich, Medical Faculty, 8032 Zurich, Switzerland. [8] University College London, Gower Street, London WC1E 6BT, UK. [9] Center for Beta Cell Therapy in Diabetes, Brussels Free University, Brussels, Belgium. ✉email: stephanie.claudinot@unil.ch; yann.barrandon@epfl.ch

The tumor protein 63 (Tp63) is a member of the Tp53 family of transcription factors and two main isoforms (TAp63 and ΔNp63) are encoded from alternative promoters[1]. Tp63 is expressed in the apical ectodermal ridge, in the epidermis and its derivatives (hair follicles, breast) and other squamous epithelia (epithelia lining the ocular surface, the buccal cavity and the esophagus), in developing teeth, in trachea, the urogenital tract, prostate, and thymus[1–5] (Supplementary Table 1). Loss-of-function of Tp63 in mouse models results in major developmental abnormalities including a hypoplastic thymus, an absence of limbs and teeth, and an absence of epidermis and its derivatives (hair follicles, mammary glands, lacrymal, and salivary glands)[6,7]. Consequently, Tp63 is considered a master gene for the development of epidermis and its appendages[8] and essential for the proliferative potential of stem cells in stratified epithelia[4,6].

The fate of an adult stem cell is determined during embryonic life and its potency is restricted to distinct tissue lineages in response to a specific microenvironment (the niche) and it is thought that lineage restriction is maintained for life[9,10]. However, in some circumstances, for instance, after chronic microenvironmental aggressions, epithelial cells may undergo a phenomenon termed metaplasia in which they adopt a different fate, e.g. squamous metaplasia in the trachea[11]. The formation of hair follicles, a landmark of mammals, requires complex mesenchymal–epithelial interactions[12,13] and it is commonly believed that embryonic epidermal cells are the only cells of the body that can respond to hair follicle morphogenetic signals in vivo[14]. Nevertheless, the corneal epithelium of an adult rabbit can form hair follicles when exposed to mouse embryonic dermis in a short-term assay[15], epidermis and hair follicles can form in the cornea of mice deficient for DKK2[16] and the corneal epithelium can convert into epidermis in Notch 1 null mice[17]. Furthermore, we have demonstrated that the thymus of the rat contains clonogenic Tp63-expressing epithelial cells of endodermal origin that can behave as bona fide multipotent stem cells of hairy skin when transplanted into a developing skin microenvironment[18]. These observations together with the fact that hair follicles and dermoid cysts are observed in a variety of non-skin tissues in adult mammals (cornea, oral cavity, esophagus, thymus) (Supplementary Table 2) strongly suggest that the ability to respond to skin morphogenetic signals is not restricted to embryonic epidermal stem cells, a notion also supported by the generation of de novo hair in the adult mouse after wounding[19]. That prompt us to systematically evaluate the potency of Tp63-expressing stem/progenitor cells isolated from squamous epithelia and non-squamous epithelia (bladder, prostate) in powerful skin morphogenetic assays that we have developed[20,21]. We have demonstrated that non-hairy adult Tp63-expressing epithelia, independently of their embryonic origin, can contribute to the formation of hair follicles and sebaceous glands when transplanted into a neonate skin microenvironment, a property maintained in serial transplantation. Importantly, some epithelial cells (bladder, prostate, thymus) necessitate a passage in cell culture to respond to a skin microenvironment, supporting the notion that tissue stem/progenitor cells may broaden potency in response to cell culture, a phenomenon thought to be associated with stem cell plasticity and reprogramming[22,23].

## Results

**Tp63-expressing clonogenic cells from non-hairy epithelia have skin-forming ability.** We have systematically examined the skin-forming ability of cultured epithelial cells isolated from simple, transitional, and stratified epithelia of adult rodents using powerful skin morphogenetic assays[21] (Fig. 1a). Tissue biopsies of footpad, oral mucosa, esophagus, vagina, trachea, bladder (dome and trigone) and prostate were obtained from 6-month-old enhanced GFP (EGFP) rats (green rat SD-Tg(CAG-EGFP)4 Osb) with the exception of gut stem cells, which were isolated from 3-week-old DsRed+ (B6.Cg-Tg(CAG-DsRed*MST)1Nagy/J) and Lgr5-EGFP (B6.129P2-Lgr5tm1(cre/ERT2)Cle/J) mice. Rat cells were then cultured onto a lethally irradiated feeder layer of 3T3-J2 cells under the strict conditions used for regenerative medicine of human epidermis[24,25], whereas duodenal stem cells from mouse DsRed+ mice were cultured according to Sato et al.[26] (Supplementary Fig. 1a). These experiments confirmed that all Tp63-positive epithelia (Supplementary Table 1) contained a population of clonogenic epithelial cells that could be serially cultured[27] (Fig. 1b, Supplementary Fig. 2a, b). Importantly, the multiplying cells obtained in these cultures, irrespective of their tissue of origin, expressed ΔNp63 isoforms (Fig. 1c, Supplementary Fig. 3a), the expression of which is critical for maintenance of epidermal stem cells in vivo[4,6]. In addition, expression of these isoforms is considered as an important indicator of the presence of stem cells in cultured epithelium grafts for ex vivo regenerative medicine[25,28]. Moreover, most cultured cells had a normal rat diploid karyotype, even if some aneuploid cells could be observed. As expected, Tp63-negative stem cells of the duodenum[29] could not grow under a microenvironment designed for epidermis but could nicely expand to form mini-guts under appropriate culture conditions[26] (Supplementary Fig. 1a). Next, we investigated the behavior of these different clonogenic cells in a powerful xeno-transplant assay in which the cultured cells were exposed to morphogenetic cues present in the developing skin of a newborn mouse. Using this assay together with serial transplantations, we have previously demonstrated that the cultured progeny of a single multipotent hair follicle stem cell can contribute to the formation of functional hair follicles and sebaceous glands for years[21]. In this study, a total of 156 transplantations were performed. As predicted, cultured stem cells from ectoderm (e.g., oral cavity, footpad, sweat glands, cervical loop of the developing incisor) and non-skin related stratified squamous epithelia of other embryonic origin (e.g., from vagina and esophagus, respectively, of mesoderm and endoderm origin) nicely integrated into the developing pelage of the mouse and contributed to epidermis, functional hair follicles, and sebaceous glands for months (Supplementary Fig. 3b, c). Surprisingly, the Tp63-expressing cells cultured from the transitional epithelium of the bladder (whether from dome or trigone) and the glandular epithelium of the prostate, both of endodermal origin, integrated very nicely into the developing pelage of the mouse and contributed to the formation of epidermis, hair follicles, and sebaceous glands for months (Fig. 1d, Supplementary Fig. 3b–e). We next examined the behavior of stem cells isolated from gut (simple epithelium of endodermal origin). As expected, cultured DsRed and non-cultured Lgr5-EGFP sorted gut stem cells were unable to integrate into the developing skin and quickly disappear from the transplant (Supplementary Fig. 1a, b). Because aneuploid cells were present in some of the primary cultures, we isolated individual single cells and expanded the clones. Aneuploid clones were then selected and transplanted. None of the aneuploid EGFP clones survived more than a few days after transplantation and never permanently engrafted (Supplementary Fig. 4a, b), a situation highly reminiscent of the lack of engraftment of cloned aneuploid human keratinocytes[30]. These observations strongly indicated that only normal diploid epithelial stem cells could permanently engraft in vivo[31], a notion of paramount importance for safe stem cell therapy. EGFP rat cells were then recovered from the xeno-transplants and cultured again as described above. Importantly, these cells kept expressing ΔNp63 isoforms when examined by

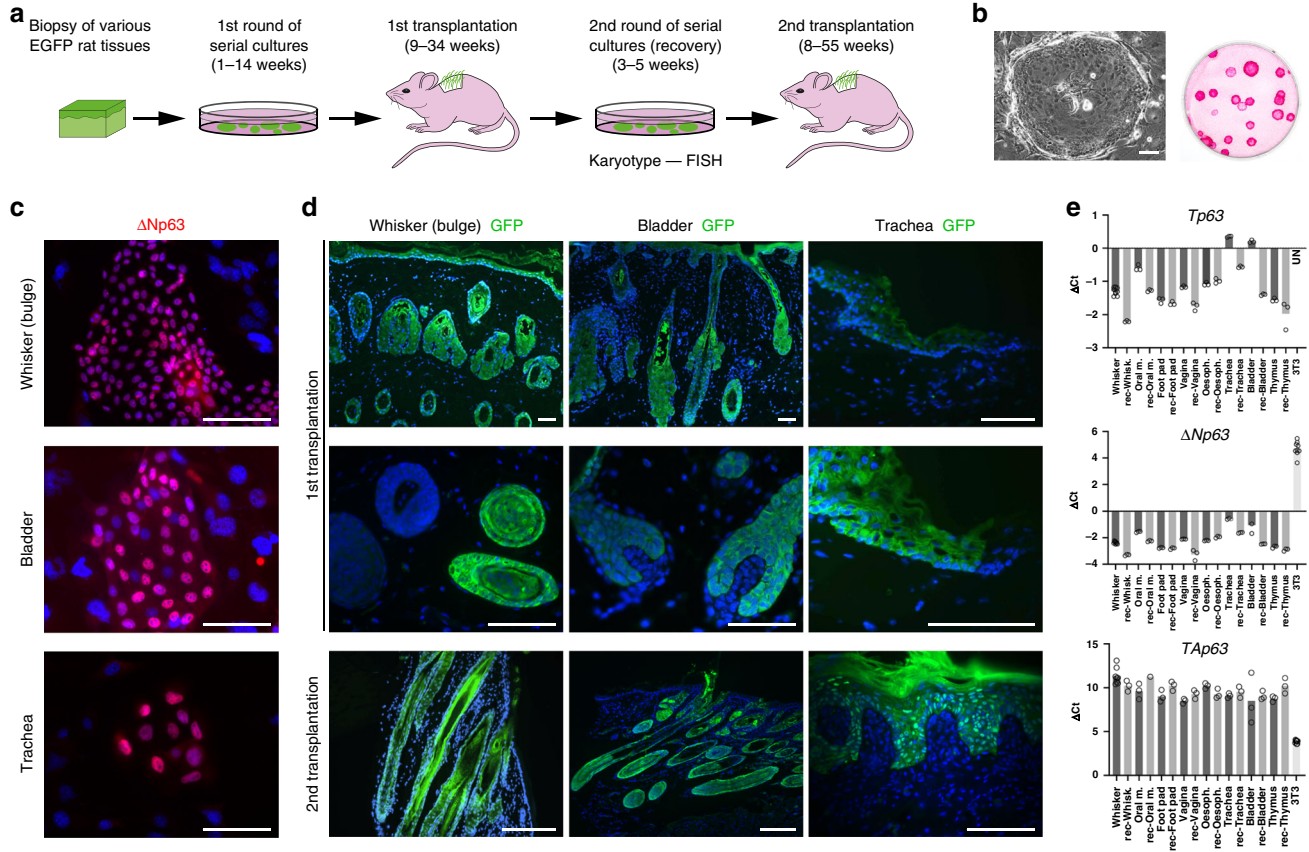

**Fig. 1 Cultured Tp63-expressing epithelial stem cells have hairy skin competence. a** Schematic representation of the experimental strategy. Various epithelial cells were obtained from EGFP rats, cultured and serially transplanted into the skin of newborn wild type mice. A typical experiment, from the initial biopsy to the end of the 2nd transplantation, usually ran for more than a year. **b** Typical appearance of colonies initiated by esophageal epithelial cells (10 independent experiments). Left: phase contrast appearance of a 7 days old progressively growing colony; bar: 100 μm. Right: appearance of 10 days old Rhodamine B stained colonies (red). **c** Immunostaining showing that clonogenic epithelial cells cultured from whisker, bladder and trachea expressed ΔNp63 (red nuclear staining). Nuclei (blue) were counterstained with Hoechst 33342. **d** Serial transplantation of Tp63-expressing epithelial stem cells from whisker, bladder, and trachea into the skin of newborn wild type mice; 34 mice were transplanted in the first round of transplantation out of which 21 displayed a long-term EGFP transplant; 23 mice were transplanted in the second round of transplantation out of which 12 displayed a long-term EGFP transplant. Representative microphotographs showing that transplanted EGFP cells from whisker and bladder formed epidermis, hair follicles, and sebaceous glands whereas tracheal cells only formed an epidermis-like epithelium. EGFP expression in the transplanted cells was revealed by immunohistochemistry. First transplantation: upper and middle panels; Upper panel: whisker: 236 days old transplant; bladder (dome): 238 days old transplant; trachea: 122 days old transplant. Middle panel: high magnification showing that the EGFP cells contributed to the differentiated layers of epidermis and hair follicles. Whisker: 100 days old transplant; bladder (dome and trigone): 238 days old transplants; trachea: 122 days old transplant. Second transplantation: lower panel; Whisker: 117 days old transplant; bladder (trigone): day 84 transplant; trachea: 100 days old transplant. Nuclei (blue) were counterstained with Hoechst 33342; bars: 100 μm. **e** Dot plot showing the expression of Tp63, ΔNp63, and TAp63 isoforms determined by qPCR analysis before and after the 1st round of transplantation. ΔCt results were normalized against housekeeping genes (*TBP*, *SDHA*, and *Tubb*). Mean of 3–9 technical replicates with individual data represented by empty circles.

immunocytochemistry and reverse transcription quantitative PCR (RT-qPCR) (Fig. 1e). Karyotype and fluorescence in situ hybridization (FISH) analyses revealed that the recovered EGFP cells had a normal rat chromosome count (*n* = 42) and had not fused with the epidermal cells of the recipient mice (Supplementary Fig. 4c). Recovered EGFP rat cells were then serially cultured before they were again transplanted into newborn mouse skin. The transplanted EGFP rat cells contributed for a second time to long-term hair follicle renewal of the mouse pelage (Fig. 1d, Supplementary Fig. 3b). The percentage of successful xeno-transplants varied from 12.25–100% (the best performers being the bladder, the sweat glands, the oral cavity, and the esophagus) compared with 75% for bona fide hair follicle stem cells (details in Supplementary Fig. 3b). Much to our surprise, the endoderm-derived tracheal epithelium, classified as pseudostratified, stood apart among all the Tp63-expressing epithelia that we examined.

As previously reported, the trachea contained a population of clonogenic cells that could grow for many passages onto a feeder layer of 3T3-J2 cells (Supplementary Fig. 2a). These clonogenic cells expressed the transcription factor ΔNp63 isoforms (Fig. 1e), *Nkx2-1*, CK-5/14 as previously reported[32,33], but did not express Involucrin (IVL) and LEKTI, markers of squamous terminal differentiation (Supplementary Fig. 5a). When transplanted, the tracheal epithelial cells never contributed to the formation of hair follicles or sebaceous glands (*n* = 8 and 17 for 1st and 2nd transplantations, respectively) but formed a squamous epithelium resembling the epidermis, a finding also observed in the second round of transplantation (Fig. 1d). This stratified squamous epithelium contained Ki67-positive basal cells and could renew for months. Moreover, it expressed some (Loricrin and LEKTI) but not all of the classical markers of squamous differentiation (e.g., IVL and Filaggrin) (Supplementary Fig. 5b). These results are

reminiscent of squamous metaplasia, a preneoplastic condition in which an epithelium repeatedly exposed to stress become squamous and stratify to resemble epidermis[34]. The morphology and layer organization of the hair follicles and sebaceous glands generated from transplanted non-hairy Tp63-expressing epithelial cells were indistinguishable from normal hair follicles or sebaceous glands (Figs. 1d and 2a). We next evaluated the expression of several transcription factors important for hair follicle development and differentiation, i.e., *Lhx2, Foxn1, Gata3, Hr,* and *Lef1*[12–14] in cells isolated from footpad, oral mucosa, esophagus,

vagina, thymus, trachea, and bladder cells recovered several months after their transplantation and contribution to murine skin. The expression of these transcription factors was upregulated in all recovered cells with the exception of the trachea in which the expression of *Lhx2, Gata3,* and *Hr* remained low (Fig. 2b). We also found that transplanted tracheal cells could only upregulate gene networks associated with squamous differentiation in accordance with their behavior upon transplantation. We next investigated the expression of SOX9, LHX2, CK-15, and CK-31 in hair follicles generated by bladder and prostatic EGFP cells by

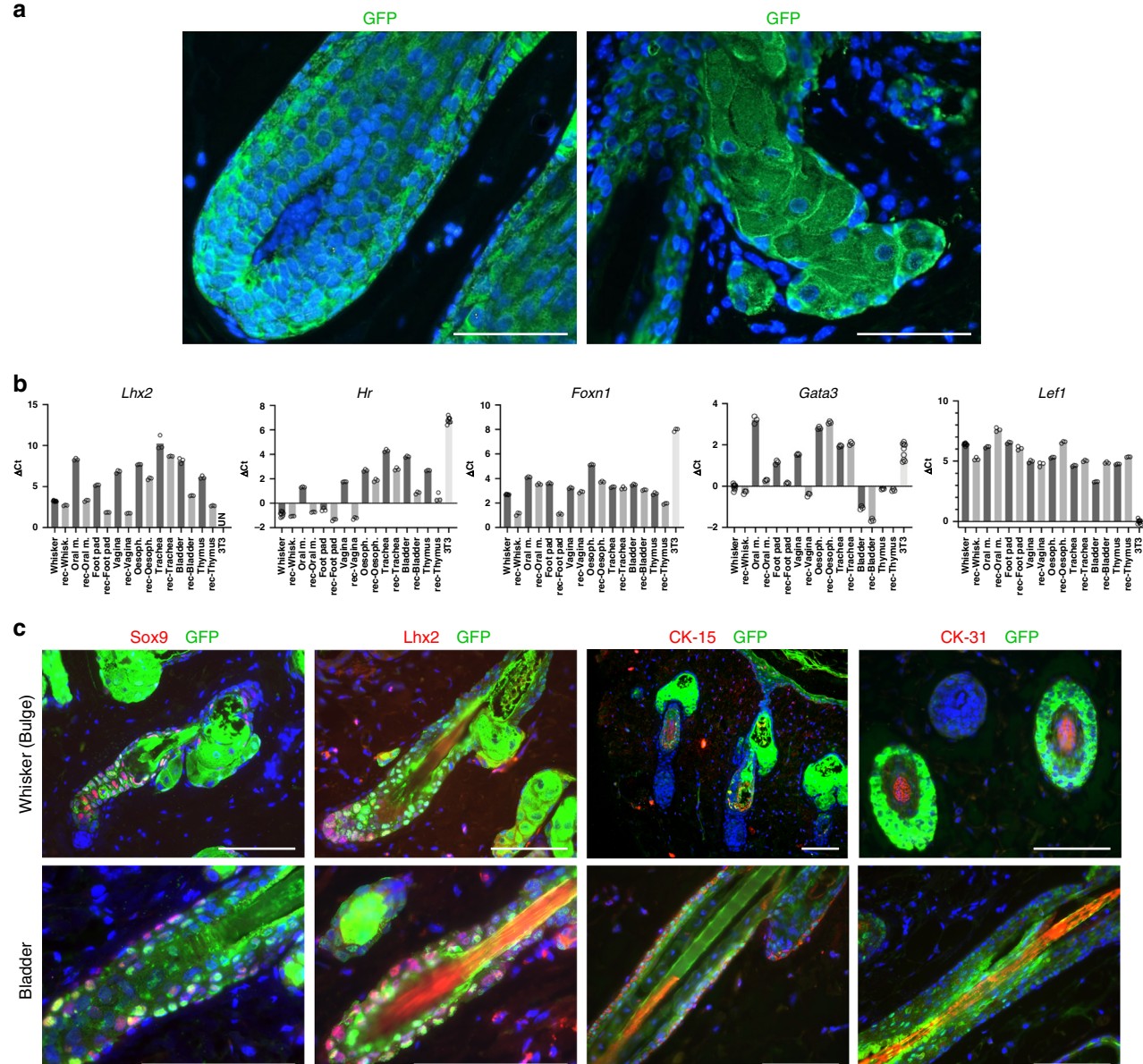

**Fig. 2 Expression of hair follicle lineage markers by transplanted bladder epithelial cells. a** Microscopic appearance of a hair follicle bulb (left) and of a sebaceous gland (right) generated by epithelial cells cultured from the bladder of an EGFP rat and transplanted into a newborn mouse skin microenvironment (103 and 238 days after transplantation, respectively), n = 3 mice. EGFP expression (green) was revealed by immunohistochemistry, nuclei (blue) were counterstained with Hoechst 33342. Note that the transplanted EGFP cells contributed to all the differentiated lineages of the hair follicle and sebaceous glands; bars: 100 μm. **b** Expression of several transcription factors important for hair follicle differentiation as determined by qPCR analysis. ΔCt results were normalized against housekeeping genes (*TBP, SDHA,* and *Tubb*). Mean of 3–9 technical replicates with individual data represented by empty circles. **c** Transplanted EGFP-positive bladder epithelial cells expressed SOX9, LHX2, cytokeratin-15 (CK-15), and cytokeratin-31 (CK-31) at the proper location in the hair follicles. For each marker, n = 3 experiments. Hair follicles and sebaceous glands generated from multipotent epithelial stem cells cultured from a whisker follicle were used as control. SOX9 (red): whisker: day 103 after transplantation, bladder: day 103; LHX2 (red): whisker: day 238, bladder: day 236; CK-15 (red): whisker: day 103, bladder: day 238; CK-31 (red): whisker: day 236, bladder: day 103. EGFP expression (green) was revealed by immunohistochemistry, nuclei (blue) were counterstained with Hoechst 33342. Bars: 100 μm.

immunocytochemistry. These proteins, linked to the hair follicle stem cell niche and hair follicle differentiation[35–37] were expressed at the right location 103 and 238 days after transplantation (Fig. 2c, Supplementary Fig. 3e), demonstrating that the bladder cells were able to turn on a transcriptional program related to hair follicle development in response to a hairy skin microenvironment. Altogether, our experiments unambiguously demonstrated that clonogenic adult Tp63-expressing stem cells, irrespective of their embryonic origin, had skin-forming ability when exposed to proper environmental cues and that they were able to behave like bona fide multipotent hair follicle stem cells as they contributed to all lineages of the hair follicle and the sebaceous glands and to hair follicle renewal for many hair cycles.

**Genetic analyses**. We next performed a microarray analysis on cells from oral mucosa, footpad, vagina, trachea, esophagus, thymus, and whisker follicle before and after their transplantation into a newborn skin microenvironment. All cells were cultured in parallel and in duplicate before they were sorted on EGFP expression and total RNA extracted. Quantitative analysis of RNA expression was then performed using the Affymetrix GeneChip Rat Expression Array 230 2.0 (31,042 gene-level probe sets) or Rat Gene ST 1.0 Array (27,342 gene-level probe sets). The results of the thymus analysis that was used for comparison were published previously and deposited at the NCBI Gene Expression Omnibus public (http://www.ncbi.nlm.nih.gov/geo) database (series record GSE21686)[18] (Supplementary Fig. 6a–d). The global gene expression in the different tissues was compared to that of the whisker hair follicle and a list of genes that were significantly up- or downregulated was established (fold-change >2; $p$ value < 0.05) (Supplementary Data 1). Only 43 genes were identified as common to oral mucosa, footpad, vagina, trachea, esophagus and thymic cells when compared with whisker follicle cells in the 1st round of culture before transplantation (Supplementary Table 3), when only four genes (*Chst11, Itgb8, Lrat, Rassf4*) remained common in the 2nd round of culture when the cells were isolated from the grafts. As expected, *Lhx2* and *Gata3* expression was lower in cells from non-hairy cells tissues before transplantation. In contrast, *Ptges* and *Ptgs1*, both implicated in prostaglandin signaling, were significantly more expressed in cells in the 1st round of culture. This observation is particularly interesting because prostaglandins interact with Wnt signaling[38], a pathway critical for hair follicle development[12], and also because they are known to inhibit wound-induced hair follicle neogenesis[39]. Altogether our results provide a possible explanation for why hair follicle formation is repressed in non-hairy Tp63-expressing cells from non-hairy tissue origin. Because the bladder epithelium was not included in our original microarray analysis, we next performed a RNA-Seq analysis on bladder (dome), trachea, vagina, thymic, and whisker follicle Tp63-expressing epithelial cells cultured before and after their transplantation into a newborn skin microenvironment. Thymic epithelial cells (PRP04MOThy1 and recovered PRP04MOThy1 #907) were from our frozen stock[18]. All cells were cultured in parallel before they were sorted on EGFP expression and total RNA extracted. Principal component analysis (PCA) and hierarchical clustering clearly indicated a switch in gene expression towards hair follicle to the exception of trachea after the cells were recovered months after transplantation (Fig. 3a, b). Because of the higher sensitivity of RNA-Seq, more genes (1182 and 629, respectively) were identified as common between the bladder (dome), trachea, vagina, and thymus when compared to whisker follicle (up- and downregulated genes, fold-change >2, adjusted $p$ value < 0.05) (Fig. 3c, listed in Supplementary Data 2 and 3). Significance of overlapping genes between comparisons was

tested pairwise using the hypergeometric distribution (phyper function in R) and was close to zero in all cases, indicating that the observed overlaps were significantly higher than expected by chance. The results of the hypergeometric test are provided in Supplementary Tables 4 and 5. Although our experiments did not formally identify those signaling pathways involved in the overriding of lineage restriction in non-hairy Tp63-expressing cells, our global transcriptional analysis and qPCR analysis of *Wnt5a, Wnt3, Wnt3a, Wnt10a, Dkk1,* and *Dkk3* expression strongly pointed to the Wnt/β-catenin/Lef-1 and Notch pathways (Fig. 3d, e). This is not surprising as there is compiling evidence that the Wnt/β-catenin/Lef-1 signaling pathway is determinant for hair follicle morphogenesis[40,41] and hair follicle neogenesis[19], and for the formation of hair follicles in the cornea of mice deficient for DKK2[16]. Furthermore, there is also evidence that a deficient Notch 1 pathway favors the conversion of the corneal epithelium into epidermis in mice[17].

**Squamous epithelia have intrinsic skin-forming ability**. To investigate if the skin-forming capability of non-skin epithelia observed in our previous experiments was acquired through cell culture, small biopsies of tissues lined by squamous (e.g., conjunctiva, cornea, footpad, esophagus), pseudostratified (trachea), transitional (bladder), and simple epithelia (gut) were obtained from adult Rosa26 mice (B6;129S-Gt(ROSA)26Sor/J). These biopsies were then transplanted together with tiny pieces of charcoal directly onto the backskin of newborn mice, at the time when hair follicles were still being formed[20]. After 4 days, the pups were killed and the zone of the skin containing a transplant, identified by the presence of charcoal, were carefully dissected out and grafted in toto onto the back of individual athymic mice (Fig. 4a). Transplants were harvested after a few weeks and examined for the presence of β-galactosidase-positive cells in hair follicles, epidermis, and sebaceous glands. We performed over 200 transplantations, the results of which are summarized in Supplementary Table 6. First, we demonstrated that corneal cells were able to form epidermis, hair follicles, and sebaceous glands confirming previous results obtained by short-term recombination of an adult central corneal rabbit epithelium with an embryonic mouse dermis[15]. Most importantly, corneal epithelial cells contributed to the formation of a new hair germ during hair cycle, thus demonstrating that they behaved like bona fide multipotent stem cells of the hair follicle (Fig. 4b). Interestingly, β-galactosidase-positive corneal mesenchymal cells (keratocytes) were never observed after transplantation and the dermal papilla was always from the recipient mouse (Fig. 4b). Second, we demonstrated that all squamous epithelia, isolated from any tissues, independently of their embryonic origin, behaved similarly and formed hair follicles, epidermis, and sebaceous glands. However, the trachea, the bladder and the thymus did not respond to skin morphogenetic signals nor did the gut (Fig. 4b, Supplementary Table 6). Although a technical issue could not formally be ruled out as a reason for the unresponsiveness of the latter epithelia, our results strongly supported the notion that stem cells of squamous epithelia stem cells have a natural competence to respond to skin morphogenetic signals. These data could explain why aberrant hair follicles and dermoid cysts are mostly observed in tissues lined by a squamous epithelium (Supplementary Table 2).

**Transplanted Tp63-expressing epithelial cells have a memory of their origin**. The transplantation of autologous stem/progenitor epithelial cells cultured from human oral mucosa to repair invalidating corneal deficiencies addresses the fundamental question of stem cell memory, transdifferentiation, and plasticity[42,43].

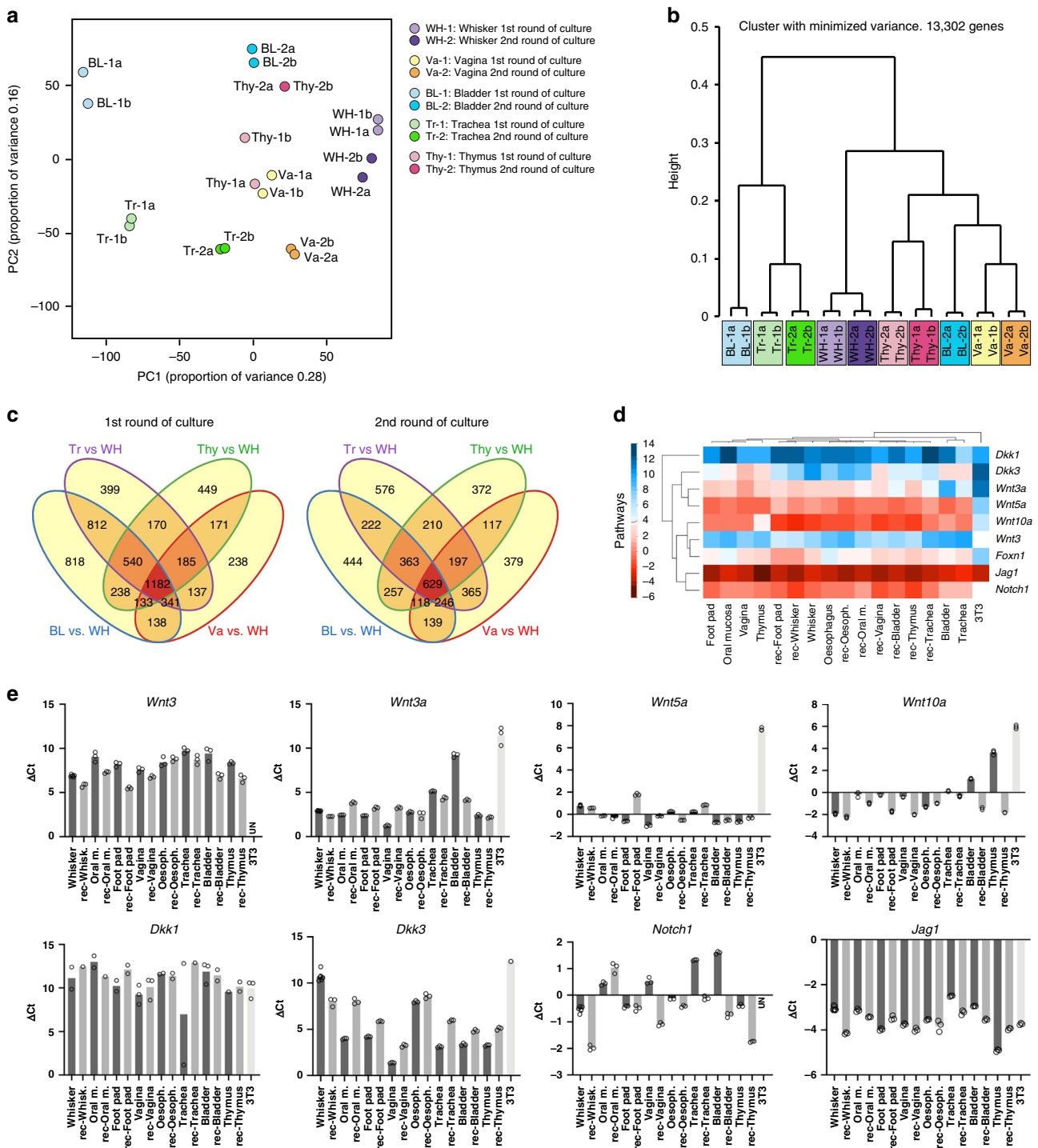

**Fig. 3 Gene expression analysis of Tp63-expressing epithelial stem cells before and after skin transplantation.** A RNA-Seq analysis was performed on epithelial stem cells cultured from whisker hair follicle (WH), vagina (Va), thymus (Thy), bladder (dome) (BL), and trachea (Tr) before they were transplanted into a newborn mouse skin microenvironment (1st round) and after they were recovered from the transplants (2nd round recovered). Cells were FACS sorted on EGFP expression before total RNA was extracted. **a** Principal component analysis (PCA) of the full transcriptomes. **b** Expression clustering analysis. **c** Venn diagram showing the number of up- and downregulated genes, fold-change >2, adjusted $p$ value < 0.05 for each tissue in comparison with hair follicle adjusted $p$ value < 0.05. **d** Heat map for the expression of some representative genes of different pathways important for hair follicle development and signaling. **e** Dot plot showing the expression of several genes of the Wnt and Notch signaling pathways in epithelial cells cultured before and after transplantation (1st and 2nd rounds of culture), determined by qPCR analysis. Housekeeping genes were used as normalizer (*TBP*, *SDHA*, and *Tubb*). Mean of 3–9 technical replicates with individual data represented by empty circles. Rec: recovered cells, 2nd round of culture.

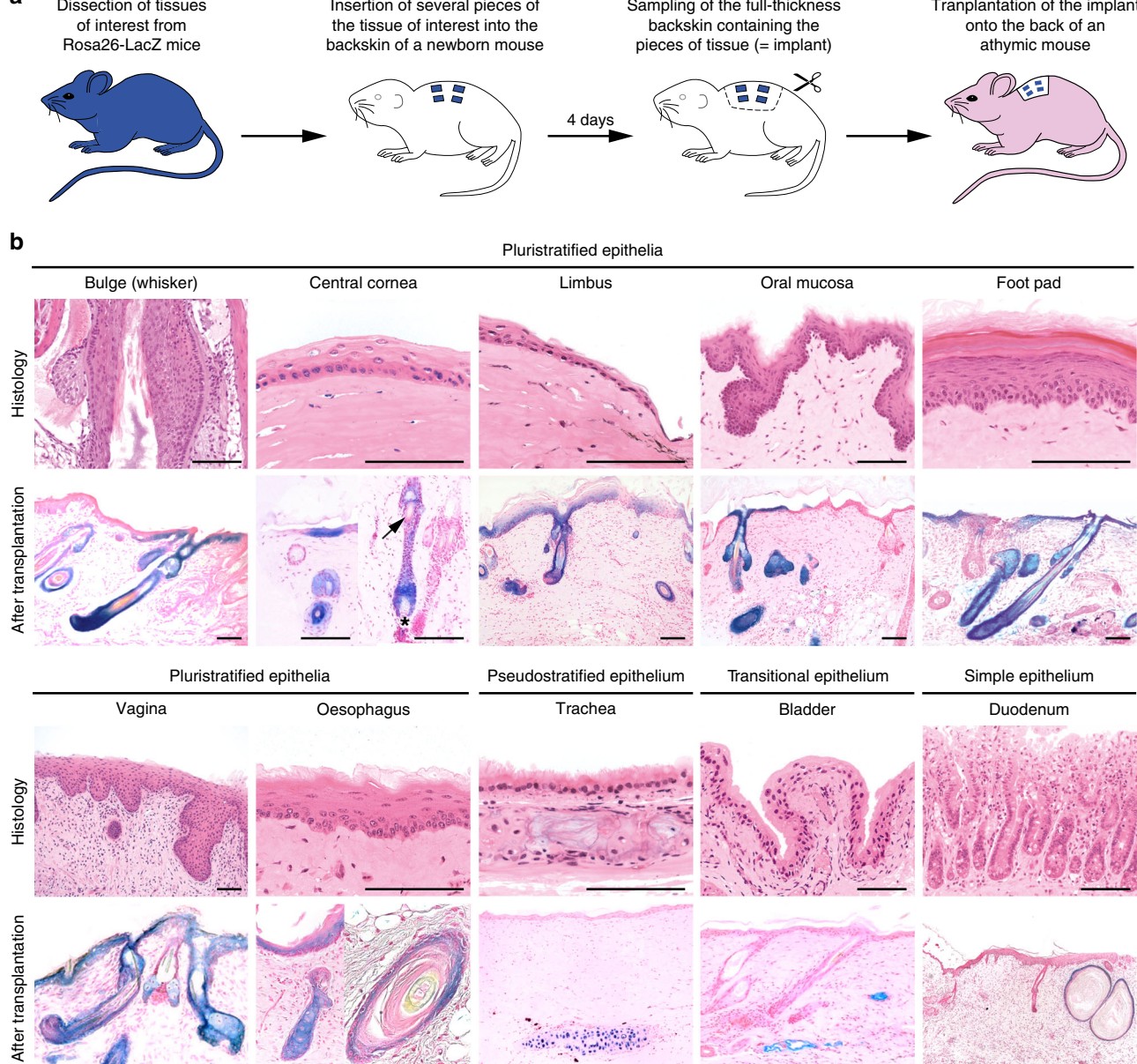

**Fig. 4 Stem cells from squamous epithelia have innate skin competence. a** Schematic representation of an experiment to determine the hair-forming ability of a non-hairy epithelial tissue. A total of 204 mice were transplanted with various β-galactosidase-expressing epithelial tissues. For each tissue, the number of transplanted mice and the number of positive transplants are given in Supplementary Table 6. **b** Microscopic appearance of the different tissues before transplantation and after transplantation into a newborn wild type mouse skin microenvironment. Donor tissue expressing β-galactosidase: blue; X-gal staining; recipient newborn mouse tissue: pink; Nuclear Fast Red. The arrow points to a club hair in a hair follicle generated from cornea; the presence of a club hair indicates that the hair follicle has cycled. The star shows a pink follicular papilla, hence indicating that it has originated from the mesenchyme of the recipient mouse skin. Bars: 100 μm.

Our transcriptional analysis also suggests that Tp63-expressing stem cells from non-hairy tissues could be easily differentiated from cultured hair follicle multipotent stem cells by the expression of specific transcription factors. For instance, *Nkx2-1* was expressed in tracheal cells, *Foxa1* in bladder cells and *Foxg1* in thymic epithelial cells but not in hair follicle multipotent stem cells. These transcription factors remained expressed in cells recovered after skin transplantation, thus identifying the recovered cells as not from hair follicle origin (Fig. 5a). In that regard, the expression of *Foxg1* was particularly informative as it was absolutely specific for thymic epithelial cells to the opposite of *Foxn1* that was expressed in many other Tp63-expressing stem cells,

including hair follicle multipotent stem cells. Hence, we examined the pattern of expression of cytokeratin 4 (CK-4), an intermediate filament protein that is expressed in some squamous epithelia, e.g., the oral mucosa, the vagina and the esophagus, but not in hairy skin or in cultured multipotent hair follicle stem cells. CK-4 expression was readily detectable in cultured epithelial stem cells isolated from vagina, oral mucosa and esophagus, but CK-4 became undetectable after the cells were transplanted into newborn mouse skin. CK-4 expression was again readily detectable in stem cells recovered from transplants, but not in the second round of transplantation (Fig. 5b). A similar pattern was observed with the expression of CK-8/18 in bladder (trigone and dome) (Fig. 5c).

**a**

| Tissue of origin | Gene symbol | 1st round of culture | | | | 2nd round of culture | | | |
|---|---|---|---|---|---|---|---|---|---|
| | | Va vs. WH | Thy vs. WH | BL vs. WH | Tr vs. WH | rec-Va vs. WH | rec-Thy vs. WH | rec-BL vs. WH | rec-Tr vs. WH |
| Vagina | *Six1* | 253.63 | 424.36 | 17.20 | 91.60 | 36.92 | 337.68 | 12.92 | 124.70 |
| | *Sox2* | 221.70 | −4.22 | NS | 221.28 | 220.93 | NS | NS | 377.17 |
| Thymus | *Isl1* | 356.14 | 643.77 | 3.52 | 563.50 | 248.25 | 814.96 | 3.08 | 447.14 |
| | *Foxg1* | NS | 811.80 | NS | NS | NS | 1605.63 | NS | NS |
| Bladder | *Foxa1* | 94.69 | 79.54 | 2551.90 | 510.41 | 30.96 | 123.41 | 1354.92 | 303.53 |
| | *Hoxd13* | 130.93 | NS | 304.91 | NS | 429.23 | NS | 218.30 | NS |
| Trachea | *Nkx2-1* | NS | NS | NS | 1400.51 | NS | 3.29 | NS | 1051.01 |

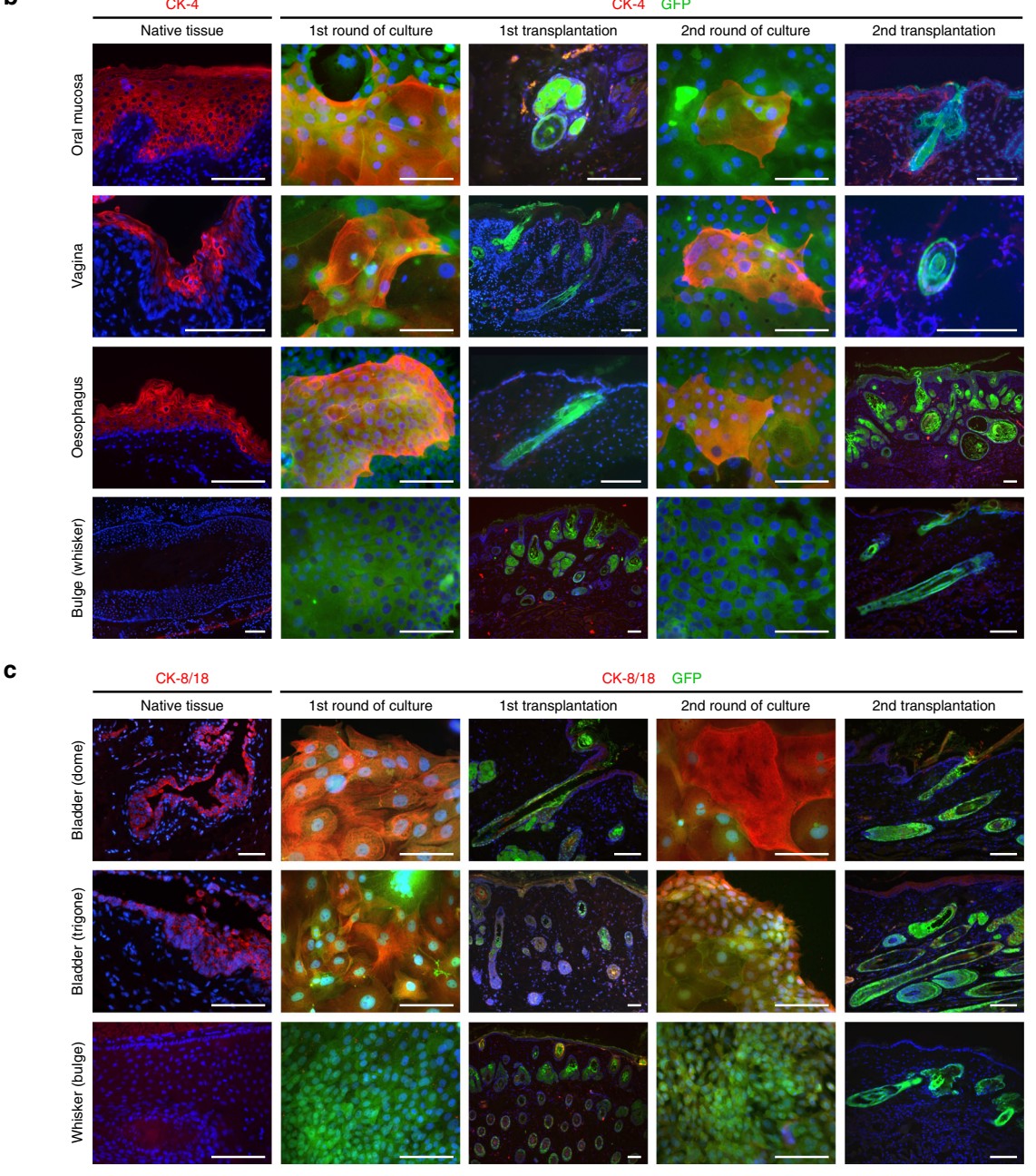

These results together with that of the global transcriptional analysis (Supplementary Data 2) demonstrated that the cells maintain a "memory" of their tissue of origin and kept expressing genes of their pre-existing program, despite functioning like bona fide hair follicle multipotent stem cells in skin transplantation assays. Previous reports had indicated that human cultured keratinocytes from the sole of the foot or palm of the hand[44,45] or the oral mucosa[43] also retained their phenotype of origin. However, in the latter reports, the original phenotype was masked in culture and revealed only after the cells were transplanted in vivo, a difference to our observations in which the original phenotype was unmasked in culture but masked after transplantation. Varying niches (newborn versus adult; healthy versus diseased) could explain this discrepancy.

**Fig. 5 Transplanted Tp63-expressing epithelial stem cells have a memory of their origin. a** RNA-Seq expression of tissue-related transcription factors for vagina, thymus, bladder, and trachea compared with hair follicle multipotent stem cells before the cells were transplanted into a newborn wild type mouse skin microenvironment (1st round) and after they were recovered from the transplants (2nd round rec); down- and upregulated genes, fold-change >2, adjusted p value < 0.05, NS: non significant. Note that the expression of these transcription factors is maintained in the 2nd round of culture after the cells have been recovered months after their transplantation. **b** Immunostaining of CK-4 (cytokeratin 4, red) and EGFP (green) in oral mucosa, esophageal, and vaginal cells; two independent experiments Oral mucosa: day 100 (1st and 2nd transplantations); vagina: day 84 and 100 (1st and 2nd transplantations); esophagus: day 99 and 100 (1st and 2nd transplantations); whisker: day 100 and 385 (1st and 2nd transplantation). Nuclei (blue) were counterstained with Hoechst 33342. Bars: 100 μm. **c** Immunostaining of CK-8/18 (red) and EGFP (green) in bladder cells; two independent experiments. Bladder (dome): day 238 and 102 (1st and 2nd transplantations); bladder (trigone): day 238 and 85 (1st and 2nd transplantations); whisker: day 236 and 385 (1st and 2nd transplantations). Bars: 100 μm. Note that the CK-4 or 8/18 keratins are expressed in culture but not in the transplants. Note that the EGFP cells recovered from the transplants re-express CK-4 or CK-8/18 when cultured. Nuclei (blue) were counterstained with Hoechst 33342.

## Discussion

Our experiments demonstrate that Tp63-expressing epithelial stem cells of non-hairy tissues and of different embryonic origin (e.g., that of cornea, esophagus, vagina, thymus, bladder, and prostate) can cross lineage boundaries when exposed to a different microenvironment (niche). In response to skin morphogenetic signals, these non-hairy unipotent/oligopotent epithelial stem cells contribute to the formation and renewal of hair follicles, sebaceous glands and epidermis for months, thus behaving like bona fide multipotent stem cells of hair skin.

Tp63-expressing epithelial stem cells can be subdivided in three groups on the basis of their response to skin morphogenetic signals. First, cells from squamous epithelia (e.g., the cornea and the conjunctiva, the epithelium lining the oral cavity, the esophagus, the vagina) can behave like bona fide hair follicle multipotent stem cells in absence of cell culture or manipulation. Second, cells from pseudostratified columnar (prostate), transitional epithelia (bladder) or thymus can also generate epidermal appendages but only after being primed by cell culture. Third, cells from trachea are the most restricted and can only form an epidermis-like epithelium (squamous metaplasia) and only after being stressed by cell culture (Fig. 6a, b). Our experiments also demonstrate that lineage restriction is not immutable and support the notion that Tp63 confers stem cells, independently of their embryonic origin, a latent hair skin competence that can be unmasked by cell culture and exposure to the microenvironment provided by a newborn hairy skin niche. The increase in competence from oligopotency to multipotency in response to a substantial change in the microenvironment (non-hairy to hairy) while maintaining the pre-existing expression of important developmentally regulated genes (e.g., Foxg1 in thymus or Foxa1 in bladder) is supportive of multilineage priming rather than reprogramming[46,47]. Hence, the key for maintenance of the functional specificity of adult Tp63-expressing stem cells resides in the microenvironment provided by stromal/mesenchymal cells as suggested by seminal mesenchymal–epithelial recombination experiments[48,49] or transplantation of sweat gland epithelial progenitors into a hormonally induced mammary gland microenvironment during pregnancy and lactation[50]. Current assays are not discriminative enough to unravel the diversity of the mesenchyme as they mostly investigate the capacity of the mesenchymal cells to embark into the adipocyte, chondrocyte, and bone lineages[51]. Challenging various Tp63-expressing epithelial stem cells with an array of signaling or small molecules, e.g., in high-throughput assays, may greatly facilitate the search for conditions important for the establishment and maintenance of a specific epithelial competence. Nevertheless, our transcriptional analysis supports the notion that the response of non-hairy Tp63-expressing epithelial cells to a newborn skin microenvironment involves the Wnt/β-catenin/Lef-1 signaling, a pathway of paramount importance for hair follicle morphogenesis[40,41,52–54] and neogenesis[19]. The involvement of Wnt signaling is further supported by pilot experiments showing that EGFP-positive oral mucosal or hair follicle multipotent stem cells do not form hair follicles when transplanted into Lef-1 deficient mice newborn skin (B6-Lef1[tm1Rug40]). In conclusion, our results demonstrate that lineage restriction is not immutable and support the notion that all Tp63-expressing epithelial stem cells (e.g., from cornea, esophagus, thymus, bladder, and prostate), have latent skin competence that can be unmasked by exposition to a newborn hairy skin niche. They also explain why aberrant hair follicles are sometimes observed in non-skin tissues in human and point to alternate sources of epithelial stem cells for reconstruction of hairy skin.

## Methods

**Animals.** Experimentations on animals were authorized by the Veterinary Commission of the Canton de Vaud (SCAV, Switzerland) (authorizations #1525, 1790, 2369, 2370, 2855, and 3294) and performed according to the Swiss legislation and the European Community Council Directive (86/609/EEC). Rats constitutively expressing Enhanced Green Fluorescent Protein (SD-Tg(CAG-EGFP)4 Osb) were from Japan SLC (Hamamatsu, Japan)[55]. Athymic (Swiss Nude[−/−]) and OF1 mice were from Charles River Breeding Laboratories (Les Oncins, France). Rosa26 (B6;129S-Gt(ROSA)26Sor/J); JAX 002073[56] and Lgr5-EGFP (B6.129P2-Lgr5[tm1(cre/ERT2)Cle]/J); JAX 008875[57] mice were from the Jackson Laboratories (USA). DsRed (B6.Cg-Tg(CAG-DsRed*MST)1Nagy/J), JAX 006051[58] mice were a kind gift from Cathrin Brisken (EPFL). B6-Lef1[tm1Rug40] mice were a generous gift from Rudolf Grosschedl, Max Planck Institute for Immunology and Epigenetics, Freiburg, Germany. Mice and rats were bred and maintained in a relative humidity 30–50%, 23 ± 2 °C, and in 12 h dark/light cycles with ad libitum water and food at the CHUV and EPFL SPF (Specific Pathogen Free) and Conventional animal facilities. Males and females were used, except for transplantations that were performed onto female nude mice. Adult mice were killed by an intraperitoneal injection of pentobarbital and rats by $CO_2$ asphyxiation; OF1 and Lef1 pups (<postnatal day 5; PN5) were decapitated.

**Tissue dissection and cell dissociation.** All tissues were dissected in cold Dulbecco Modified Eagle's Medium (DMEM, Gibco-Invitrogen). Tissues other than gut were obtained from EGFP rats at postnatal ages PN9 (sweat gland), PN57 (whisker bulge), PN200 (oral mucosa, footpad, vagina, trachea, esophagus) and PN201 (dome and trigone of bladder, prostate, and ureter) and were dissociated at 37 °C in a solution of 0.05% trypsin (MP Biomedicals, Inc.) – 0.1% ethylenediaminetetraacetic acid (EDTA) (Sigma). OF1 and Lef1 mouse backskin were from pups (PN0-2), whereas Rosa26-lacZ tissues were obtained from adult mice (>PN21). Duodenal crypts were obtained from Lgr5-EGFP or DsRed adult mice (>PN21). After cleaning the duodenum with cold 1× Dulbecco-PBS medium (Gibco), crypts were isolated by a 20 mM EDTA treatment and filtration through a 70 μm cell-strainer (Beckton Dickinson). Crypts were then digested with a solution containing TrypLE (Life Technologies), 500 μM N-acetyl-L-cysteine (Sigma Aldrich), 10 mM Rho kinase, 1 mg mL[−1] DNaseI (Roche). Lgr5-EGFP cells were then sorted with an Aria-2 FACS sorter (Beckton Dickinson).

**Cell culture.** Mouse embryonic 3T3-J2 fibroblasts, a generous gift from Howard Green (Harvard Medical School, USA), were cultured in DMEM supplemented with 10% calf serum (Sigma) and lethally irradiated at 60 Gy before they were used as feeders[59]. Rat epithelial cells were cultivated on a feeder layer of lethally irradiated 3T3-J2 cells in a 3:1 mixture of DMEM and Ham's F12, (Gibco-Invitrogen), supplemented by 10% fetal calf serum (FCS, Gibco), 10[−6] M cholera toxin (Sigma), 2.10[−9] M 3,3′,5-triiodo-l-thyronin (T3) (Sigma), 5 μg mL[−1] insulin (Sigma) and 0.4 μg mL[−1] hydrocortisone (Calbiochem)[20,21]. Cells were fed every 3–4 days and 1 ng mL[−1] of human recombinant EGF (QED Biosciences) was added at the first feeding. Cells were then serially passaged once a week. Single-cell isolation was performed under a Zeiss inverted microscope with a ×10 phase objective under

**a**

| Tissue | Epithelial type | Embryonic origin | In vivo | | Hairy skin morphogenesis | | |
|---|---|---|---|---|---|---|---|
| | | | Tp63 | Aberrant skin appendages | Non-cultured cells | Cultured cells | |
| Whisker | Stratified squamous keratinized | Ectoderm | + | + | + | + | 1 |
| Foot pad | Stratified squamous keratinized | Ectoderm | + | + | + | + | |
| Sweat glands | Simple and stratified cuboidal | Ectoderm | + | No data | No data | + | |
| Oral mucosa | Stratified squamous non-keratinized | Ectoderm | + | + | + | + | |
| Vagina | Stratified squamous non-keratinized | Mesoderm | + | + | + | + | |
| Oesophagus | Stratified squamous non-keratinized | Endoderm | + | + | + | + | |
| Thymus | Reticular | Endoderm | + | + | − | + | 2 |
| Bladder | Transitional | Endoderm | + | No data | − | + | |
| Prostate | Pseudostratified columnar | Endoderm | + | No data | No data | + | |
| Trachea | Pseudostratified | Endoderm | + | No data | − | Squamous | 3 |
| Ureter | Transitional | Mesoderm | + | No data | No data | Cysts | |
| Duodenum | Simple | Endoderm | − | − | − | − | |

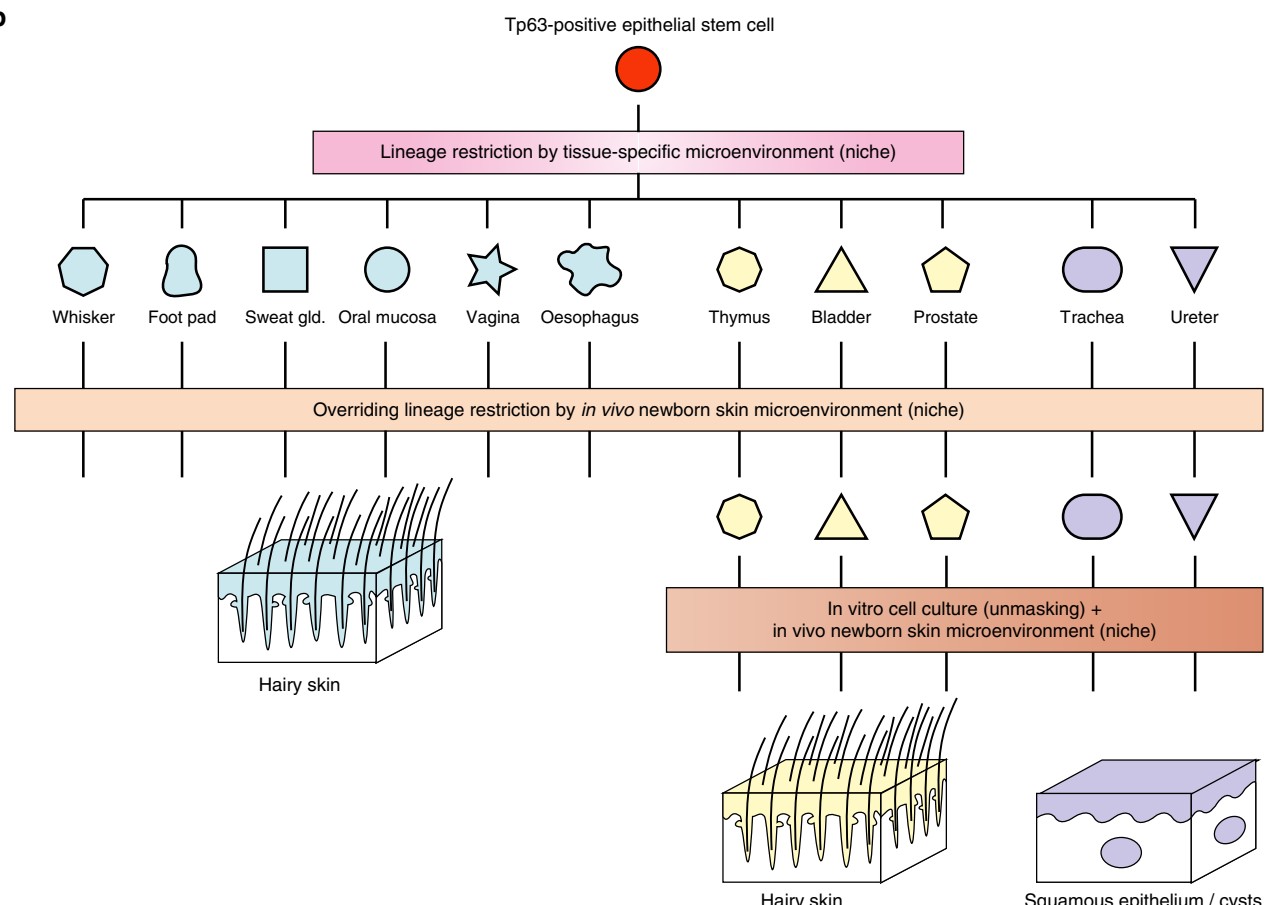

**Fig. 6 Tp63-expressing epithelial stem cells can cross lineage boundaries when exposed to a hairy skin microenvironment. a** Tp63-expressing epithelial stem cells can be classified in three groups depending on their response to skin morphogenetic signals. This response is independent of the embryonic origin. Group 1 includes stem cells that can behave like bona fide hair follicle multipotent stem cells in absence of manipulation (cell culture). This may explain why the formation of aberrant hair follicles and of dermoid cysts is mostly observed in tissues lined by a squamous epithelium. Group 2 includes stem cells that can generate epidermal appendages but only after cell culture (needs unmasking). Group 3 includes stem cells that can only form an epidermis-like squamous epithelium (squamous metaplasia) in response to cell culture. **b** A microenvironment specific for each tissue determines the fate of a Tp63-expressing epithelial stem cell and it can be overridden by a hairy skin microenvironment.

laminar flow hood. Single cells were individually aspirated into a sterile elongated Pasteur pipet and inoculated into individual culture dishes already containing lethally irradiated 3T3-J2 cells[60]. Cells were then cultured as above; clones were passaged after 7–10 days and expanded. Mouse crypt cells were cultured according to Sato's protocol[26]. In brief, dissociated cells were suspended in Matrigel (BD Bioscience) and cultured in a crypt medium containing 1× Glutamax (Invitrogen), 10 mM HEPES (Gibco), 1× penicillin–streptomycin (Invitrogen), 1× B-27 (Invitrogen), 1× N-2 supplement (Invitrogen), 1.25 mM N-acetyl-L-cysteine (Sigma Aldrich), 50 ng mL$^{-1}$ EGF (QED Bioscience), 100 ng mL$^{-1}$ Murine Noggin (Peprotech), 1 µg mL$^{-1}$ Recombinant Mouse R-spondin1 (R&D Systems) and 10 µM Y-27632 (Calbiochem) in Advanced DMEM-F12 (Invitrogen). After 7 days in culture, crypts were dissociated into a single-cell suspension with a 0.05% trypsin–0.1% EDTA solution before being transplanted.

**Karyotype and FISH.** Chrombios GmbH (Raubling, Germany) performed the karyotype and FISH analyses. Cells were cultured as described above for 5 days before they were incubated with 0.2 µg mL$^{-1}$ colcemid for 6 h. Cells were then dissociated, treated with 75 mM KCl and fixated with a 3:1 acetone–methanol solution at −20 °C. For karyotype and chromosome counting, slides were stained with DAPI and analyzed with a Zeiss Axioplan II microscope equipped with a b/w CCD camera. Images were snapped using the Smartcapture VP software (Digital Scientific, Cambridge, UK). The image processing software Quips (Vysis, USA) was used for reversed DAPI banding. For FISH analysis, rat genomic DNA was labeled by DOP-PCR with FITC-dUTP (green) and mouse satellite DNA with TAMRA-dUTP (red) or inversely. Analysis and superposition were done with the Smart-Capture VP software (Digital Scientific, Cambridge).

**Transplantation.** Non-cultured epithelial tissues were transplanted according to our published protocol[20]. In brief, small full-thickness biopsies of the epithelial tissues of interest were obtained from Rosa26 adult mice and implanted onto the backskin of OF1 mice (PN0-PN2). Sterile charcoal was placed next to the transplant to facilitate its future localization. After 3–4 days, OF1 the pups were killed and the area of skin containing a graft identified by the presence of charcoal was dissected and transplanted in toto onto the back of adult athymic mice. Transplants were harvested after 3–4 weeks and treated for histology.

Cultured epithelial cells or fresh gut cells were transplanted according to our published protocol[21]. In brief, a small piece of full-thickness skin was dissected from the back of OF1 newborn pups (PN0–PN1) and incubated in a 2% EDTA-1× PBS solution for 150 min at 37 °C and then washed in DMEM. The epidermis was then gently separated from dermis with a 30-gauge needle in two separate places to create small pockets. In total, $3 × 10^5$ to $5 × 10^5$ cells suspended in culture medium were then carefully injected in each pocket using a 30-gauge needle and allowed to attach for 1 h at room temperature and then maintained at 4 °C overnight. A skin flap was then made on the back of each recipient athymic mouse. Samples were then individually transplanted and stitched in place onto the back of athymic mice (one sample per mouse), with the dermis of the transplant facing the thoracic fascia of the athymic mouse. To prevent drying, transplants were then quickly covered with the flap which edges were carefully stitched with 6/0 sutures (Ethilon, Johnson and Johnson). After 2 weeks, the flap was opened and the transplant air exposed. Transplants were then kept in place for several months before they were harvested. Cultures were usually used between passage I to V, except for control bulge cells of the whisker (YR219P5cl7) that maintained a normal karyotype up to passage XI–XIV.

**Histological procedures.** Cultured cells were grown on glass coverslips (Thermo Scientific) for 4–7 days before they were fixed with 4% paraformaldehyde or 3.7% formaldehyde depending on which antibody was later used. Biopsies or transplants were fixed with 4% paraformaldehyde or 3.7% formaldehyde before they were included in paraffin or cryopreserved. Five µm paraffin sections were performed using a Leica RM2265 microtome and were either stained with hematoxyline–eosin or used for immunohistochemistry. For frozen sections, unfixed tissues were first incubated in 30% sucrose in PBS 4 °C overnight, then embedded in optical coherence tomography compound (Sakura). Eight µm cryosections were then obtained on a Leica CM3000 cryostat and post-fixed with either 4% paraformaldehyde or 3.7% formaldehyde depending on which antibody was later used. To reveal β-galactosidase activity, transplants were fixed with 4% paraformaldehyde and incubated in X-Gal solution (4 mM potassium ferrocyanide, 4 mM potassium ferricyanide, 1 mg mL$^{-1}$ X-GAL (Eurobio), 2 mM MgCl$_2$ in PBS). Samples were then embedded in Historesin (Leica), 4 µm sections were obtained with a Leica 2065 microtome and nuclei were counterstained with Fast Red (C.I. no. 20760, Fluka). For immunochemistry, deparaffinized sections, cryosections, and fixed cultured cells were incubated with the primary antibodies of interest. Secondary antibodies were coupled with Alexa488 or Alexa568, before the nuclei were counterstained with Hoechst 33432 (Sigma). The different antibodies, their working dilutions and manufacturers are listed in Supplementary Table 7. Samples were then observed on a Zeiss Axioskop coupled to an Axiocam MRc camera or on a Zeiss Axioplan2 coupled to an Axiocam MRm camera. Alternatively, they were observed on an Axio Imager Z1 microscope coupled with an Axiocam 506 mono camera (Zeiss). Pictures were captured with the Zeiss Axiovision Release 4.6.3 or

Zeiss Efficient Navigation software. The time of exposure was adjusted to have readable EGFP levels in hair follicles and consequently the high EGFP-expressing sebaceous glands were often overexposed. False colors were added on imaged captured in black and white with the ImageJ software. Hoechst intensity was normalized with Adobe Photoshop or ImageJ.

**Microarray analysis.** A microarray analysis was performed on Tp63-expressing epithelial cells isolated from the whisker, footpad, oral mucosa, esophagus, vagina, trachea, and thymus of EGFP rats[55]. All cells were cultured in parallel and in duplicate before they were sorted on EGFP expression. Two sets of fluorescence-activated cell sorting (FACS)-sorted cells were used in the analysis; the first set consisted of cells cultured before they were transplanted into a newborn skin microenvironment (before transplantation), whereas the second set consisted of the same cells cultured after they were recovered from the transplants (after transplantation). Cells were dissociated with trypsin/EDTA, filtered through a 70 µm cell-strainer (Becton Dickinson), centrifuged before they were suspended in 1× HBSS—2% FCS—20 mM Hepes—2 µg mL$^{-1}$ DAPI (Sigma). EGFP cells were then sorted using a FACS DIVA or Aria-2 (Becton Dickinson). Total RNA was then extracted from the sorted cells using the RNeasy Mini Kit (Qiagen) and the quality of the RNAs was measured using the Agilent Bioanalyzation system (Agilent Technologies). Quantitative analysis of RNA expression was then performed using Affymetrix gene chip cDNA microarrays (Affymetrix). cDNA synthesis, hybridization to the Affymetrix GeneChip Rat Expression Array 230 2.0 (31,042 gene-level probe sets) or Rat Gene ST 1.0 Array (27,342 gene-level probe sets) and analysis was performed by the Lausanne Genomic Technologies facility using standard protocols. P values were calculated using Bioconductor limma package[61] to identify differentially expressed genes from each group and probe sets with a false discovery rate[62], p values < 0.05 were considered significant. The results of the thymus analysis were published previously and could be found on the NCBI Gene Expression Omnibus public (http://www.ncbi.nlm.nih.gov/geo) database (series record GSE21686)[18]. The analysis and clustering were performed with a DNA-Chip Analyzer (dChip) software (http://biosun1.harvard.edu/~cli/dchip.exe). A dChip model-based expression with a "perfect-match only" approach was used. The list of all genes specifically upregulated in hair follicle versus all other non-transplanted tissues (fold-change = 1.5) was used to define a "hair follicle blueprint". The expression of these "blueprint" genes was then analyzed in cells recovered after transplantation.

**RNA-Seq analysis.** A RNA-Seq analysis was performed on Tp63-expressing epithelial cells isolated from whisker, bladder, vagina, trachea, and thymus from EGFP rats[55]. Thymic epithelial cells (PRP04MOThy1 and recovered PRP04MOThy1 #907) were from our frozen stock[18]. All cells were cultured in parallel before they were sorted on EGFP expression as described above in the microarray section. Two sets of FACS-sorted cells were used in the analysis (Supplementary Fig. 7); the first set consisted of cells cultured before they were transplanted into a newborn skin microenvironment (before transplantation), while the second set consisted of the same cells cultured after they were recovered from the transplants (after transplantation). Total RNA was extracted from the sorted cells using the RNA minikit (Qiagen), with addition of a DNaseI- treatment (RNase-Free DNase set, Qiagen), according to the manufacturer's instructions. RNA quantification was performed using a nanodrop spectrophotometer (Thermofisher scientific). The Genomic Technologies Facility Platform (University of Lausanne, CH) performed the RNA-seq analysis. RNA quality was assessed using a Fragment Analyzer (Advanced Analytical Technologies). All RNAs had a RQN between 7.5 and 10. The two best samples were used for sequencing. RNA-seq libraries were prepared using 400 ng of total RNA and Illumina TruSeq Stranded mRNA reagents (Illumina), on a Sciclone liquid handling robot (PerkinElmer) using a PerkinElmer-developed automated script. The resulting libraries were used to generate clusters with the TruSeq SR Cluster Kit v4 (Illumina), and sequenced using an Illumina HiSeq 2500 sequencer with TruSeq SBS Kit v4 reagents. Sequencing data were processed using the Illumina Pipeline Software version 1.82. Purity-filtered reads were adapter- and quality-trimmed with Cutadapt (v.1.8)[63]. Reads matching to ribosomal RNA sequences were removed with Fastq_screen (v. 0.9.3) and remaining reads were further filtered for low complexity with Reaper (v. 15-065)[64]. *Rattus norvegicus* genome (Assembly, v. Rnor 6.0) and STAR (v. 2.5.2b)[65] were used to aligned reads. The number of read counts per gene locus was summarized with Htseq-count (v. 0.6.1)[66], with *R. norvegicus* gene annotation (Ensembl, v. Rnor 6.0.87). Quality of the RNA-seq data alignment was assessed using RSeQC (v. 2.3.7)[67]. Statistical gene analysis was performed in R (version 3.3.2). Genes with low counts were filtered out with a minimal threshold of one count per million (cpm) in at least one sample. Library sizes were scaled using TMM normalization (EdgeR package, v. 3.14.0)[68] and log-transformed with Limma Voom (Limma package v. 3.28.21)[69]. Differential expression was analyzed with Limma[70] and a moderated t test was applied for each comparison. Genes were computed by the Benjamini-Hochberg method, controlling for false discovery rate. Genes were selected as differentially expressed with a log2 Fold-change higher or lower than 2 and an adjusted p value < 0.05. Lists of genes were then compared and analyzed by Gorilla software (http://cbl-gorilla.cs.technion.ac.il). The significance of overlap between lists of differentially expressed genes (adjusted p value ≤ 0.05, absolute fold change ≥2) before transplantation and following transplantation was calculated using the hypergeometric distribution (phyper

function in R). In brief, lists were compared pairwise by inputting the number of intersecting genes (i), number of genes in comparison 1 (m), the number of genes in comparison 2 (k) and the total number of expressed genes (N) into the phyper function in R and p values calculated for depletion and enrichment. Venn diagrams were created using the Vennerable package (version 3) in R.

**RT-qPCR.** For quantitative RT-PCR, one µg of total RNA from FACS-sorted cultured cells was reverse transcribed using the superscript III reverse transcriptase kit (Invitrogen). Technical triplicates of cDNAs were amplified using the Taqman Universal Mastermix II, no UNG (Applied Biosystem) on a QuantStudio 6 Flex Real-Time PCR System (Thermofisher). Data analysis was performed using the Expression Suite Software (Applied Biosystems) using TBP, SDHA and Tubb as normalizers. $\Delta$Ct expression (Curve threshold of gene of interest−curve threshold of housekeeping genes) was reported in GraphPad Prism v7 and 8.4.3. Primers (Applied Biosystems) are listed in Supplementary Table 8.

**Reporting summary**. Further information on research design is available in the Nature Research Reporting Summary linked to this article.

## Data availability

The data sets generated during the current study are available from the corresponding authors on reasonable request. The microarrays and RNA-Seq data are accessible from NCBI's Gene Expression Omnibus (GEO accession numbers GSE116717 and GSE116719, respectively).

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

## Acknowledgements

We are grateful to Austin Smith and Shahragim Tajbakhsh for stimulating discussions, to Ornella Barrandon for constructive comments and proofreading the manuscript. We are thankful to Liliane Schnell, Louis Mercier, Olga de Souza and Dorinne Savoy for expert help with histology. We are also thankful to Miguel Garcia, Valérie Glutz and Loïc Tauzin from the EPFL Flow Cytometry Core Facility, Bastien Mangeat from the EPFL Gene Expression Core Facility, Keith Harsman, Otto Hagenbüchle, Sylvain Pradervand, Johann Weber, Karolina Bojkowska, Leonore Wigger and Sandra Calderon from the Genomic Technologies Facility at the University of Lausanne (UNIL) for their expert support. We also thank Mark Ibberson from the Swiss Institute of Bioinformatics, Vital-IT and UNIL for his expert help with the RNA-Seq analysis. We thank Spenson Watson, Joanna Joyce and Declan Lunny for the SOX9 and CK-15 antibodies, respectively. The work was supported by the EPFL, the CHUV and by grants from the Swiss National Science Foundation (PNR46 4046-101111 to A.R., 3100A0-104160 to Y.B., 31003A-118332 to T.M. and Y.B.) and from the European Economic Community through the 6th (EuroStemCell) and 7th (EuroSyStem and OptiStem) framework programs to A.R. and Y.B. The work was also supported by the Lee Seng Teik and Lee Hoo Leng Distinguished Professorship in Plastic Surgery and Regenerative Medicine at Duke-NUS Medical School and by the Institute of Medical Biology A*STAR, Singapore.

## Author contributions

S.C., J.S., H.O., C.G., D.L., M.N., T.M. and A.R. performed experiments. P.B. and G.M. helped with the interpretation of the genome-wide experiments. S.C., A.R. and Y.B. assisted in the design of the experiments, in the interpretation of the results and wrote the paper.

## Competing interests

The authors declare no competing interests.
