## [Peer Review File · Nature Communications]

Reviewers' comments:

Reviewer #1 (Remarks to the Author):

In this manuscript, Claudinot et al., studied the plasticity of p63-expressing epithelial cells derived from various tissues including all three germ lines (ectoderm, endoderm and mesoderm). The authors transplanted (with and without culturing) p63-expressing cells from various tissues onto back skin. The authors further claim that the transplanted cells survived and contributed to hair follicles and epidermis. Interestingly, the authors found that transplantation of p63-expressing cells from central cornea, limbus, oral mucosa, foot pad, vagina, esophagus, bladder and whisker bulge but not the cells isolated from trachea. The authors also claim that certain keratin genes that are characteristics to tissue of origin are gradually lost and they instead acquired the markers of host tissue. Furthermore, the authors performed RNA-seq analysis on cells isolated from first and second round of transplantation and directly compared them to that of cells before transplantation and found that transcriptomes of transplanted cells are more similar to that of skin hair follicles and the epidermis while losing the characteristics of tissue of origin. Interestingly, they found that p63-expressing cells derived from tissues such as trachea did not contribute to hair follicle regeneration. Instead, the transplanted cells formed contributed to stratified squamous epithelium of the interfollicular epidermis. These findings are of significant interest in milieu of understanding the mechanisms of cellular plasticity and harnessing the potential of plasticity for cell-based therapies. However, to further strengthen these findings the authors should address the following concerns.

1. Based on the presence of labeled transplanted cells in hair follicles the authors state that cells isolated from various organs have the ability to generate hair follicles and interfollicular epidermis. However, the presence of labeled transplanted cells is a mere indication of cells filling the gaps in the damaged tissue. Therefore, the authors should check markers of various compartments of the hair follicles. For example, the authors should check the expression of SOX9, KRT15, LGR5, LHX2 and GLI1 to determine whether p63-expressing cells derived from other tissues generate/contribute to appropriate cell types in different compartments of the hair follicle?
2. In Figure 3b, p63-expressing cells derived from only pluristratified epithelia contribute to hair follicles but not from cells derived from pseudostratified, transitional or simple epithelium. The authors should check whether the markers of donor tissue continue to express when they do not generate/contribute to hair follicles. Such analysis would aid in better understanding why certain p63-expressing cells from some tissues do not exhibit plasticity.
3. Along the similar lines, the authors should utilize their RNA-seq data to map the differences in p63-expressing cells derived from trachea vs other pluristratified organs. Although the functional validation of potential candidates is not in the scope of the current manuscript, mapping the differences in transcriptome will aid in understanding the mechanisms of plasticity in future studies.
4. In figure 5a, the authors state that developmental origin of vaginal squamous epithelium is mesoderm. However, it should be noted that distinct parts of the vaginal epithelium are derived from endoderm or mesoderm during development. Therefore, the authors should specifically state which part of the vaginal epithelium was used to isolate p63-expressing cells.
5. Plasticity associated with cell transplantation or cell culture has been studied in many tissues. More specifically, epithelial cells isolated from sweat glands have been shown to acquire the characteristics of mammary epithelial cells when the host animal undergoes microenvironmental changes due to pregnancy and lactation. Such plasticity was not observed in normal hosts. The current findings are similar to prior work (Lu et al., Cell, 2012). It is important to acknowledge these prior findings.

Reviewer #2 (Remarks to the Author):

The authors conduct a comprehensive analysis of the capacity of Tp63 expressing cells to produce hairy skin when placed in the appropriate niche. This is an interesting and important story. The volume of in vivo studies is impressive, however the analysis of the extensive but disparate global gene expression data requires significant revision in terms of presentation and downstream analysis that will improve manuscript significantly.

Integrating data analysis from multiple array platforms and RNA-seq is challenging. Details of how these were compared are not given. Please clarify in both text and in detail in methods? Was analysis limited to comparison of samples on single platform and then comparison of differential lists?

In figure 2C what is the significance of overlaps?

The authors mentions their pathways analysis did not detect WNT pathways. What did this identify? The results are not illustrated or described? Did they identify p63 enrichment of p63 targets?

Moreover they then show what I believe to be array expression of a number of individual genes in the heatmap in figure 2d but the scale is labelled "pathways"?

The authors should additionally evaluate the role of direct p63 target genes in regulating the differentially genes they identify either through overlapping with published p63 ChIP-seq data or through use of publicly available or commercial tools such as ENRICH or IPA which provide quick ways to evaluate this.

The authors refer to "Memory" of transplanted cells on page 10 line 253. This is suggestive on an epigenetic program that is maintained. Have the authors looked at changes in DNA methylation in any of these analyses? Moreover, given queries raised above relating to p63 target genes, what are the transcription factors driving expression of these maintained genes and are these also enriched for p63 targets? i.e. are the Tp63 positive cells primed for expression for squamous cells as would be suggested from extensive analysis from ChIP-seq studies that suggests Tp63 bookmarks squamous specific enhancers.

It would be intriguing to map these transcriptional trajectories onto the schematic in Figure 5b and I think it may be possible with data contained herein.

In Figure 1 the authors look at expression of TA and DN Tp63 isoforms, they should look at the alpha, beta and gamma c-terminal isoforms as they play important roles in differential target gene regulation.

Figures 1C/E why are there no error bars? Are these single PCR reactions? It appears these results were not biologically replicated? But only technically replicated? Legends should state this and include standard deviation of technical replicates. Is this more compelling than merely relying on array data and either way does this then warrant inclusion of this many panels in main figures?

Labels of these PCR panels are very hard to read and would benefit from a colour coding? This is also true for Figure 2e. perhaps the colour coding could be matched to 2a PCA plot.

The same is true for figure 2b where same colour scheme should be used for labels.

With regard to aneuploid cells and their capacity to transplant is this true when tissues are exposed to DNA damage. E.g. in UV containing skin wherein mutated clones containing for example p53 mutations may pre-exist is there a potential for transplantation to induce expansion of the potentially cancer pre-cursors?

What is the clone forming capacity in vitro (Holo/Meri/Parclones of tracheal cells compared to other cell types?)

Nomenclature for p63 should be consistent. Human p63 is TP63, Rat Tp63 and mouse is Trp63 this should be checked carefully throughout as well as same for other genes.

The authors should change "genome-wide screen" to something that reflects this is analysis of gene expression e.g. "global transcriptional analysis"

Reviewer #4 (Remarks to the Author):

In this manuscript, Claudinot et al., demonstrate contribution of epithelial cells derived from non-skin p63+ epithelia to epidermis and/or sebaceous glands and hair follicles. Furthermore, with microarray and RNA-seq analysis they show that although cells can integrate they remain a molecular memory in the form of donor tissue transcription factor expression. Also, that Wnt signaling factors are beneficially expressed, possibly facilitating, as they speculate, the conversion to hair follicle and epidermal cells. Finally, they show that the default integration capacity depends on epithelial classification, and that cell culture conditions can unmask the potential of non-squamous epithelial cells to integrate with epidermis and hair follicles. Overall, this is an interesting study with the curious observation that non-hairy epithelial cells have the ability to integrate into hair and/or epidermis. However, two major concerns regarding the considerably loose interpretation of potential p63 functions and stem cell behavior dampen my enthusiasm.

Major concerns:

1. A major concern is that the authors invoke stem cell functions and translational application prospects which is a big stretch, although they were otherwise careful with wording when they describe the grafting success as that cultured cells "contribute" or "nicely integrated". It appears that percent of integrations refers success of integration at any detectable level, but does not reflect percent of full formation potential. As remarkable as "contribution" and "integration" of non-skin epithelial cells is, it does not at all prove competence. From the images provided in Fig. 1d for bladder and Fig S3b for vagina and prostate, for example, most parts of ORS and infundibulum appear unlabeled. For the bladder cells, the hair follicle structures can be barely recognized and some larger form of growing masses are seen instead. Unsurprisingly, the non-hairy, but footpad skin-derived cells performed much better. As long as it is not 100% of grafted cells within a given larger area of epidermis, or the entire epithelial complement of single hair follicles (bulge, matrix, all diff lineages), competence to induce and/or maintain the tissue is not established. As such stem cell functions implied for the grafted epithelial cells are not established. The authors have previously shown in Ref 32 remarkable competence of transplanted bulge to supply the entire follicle unit in successive cycles. Nothing less should be employed to show competence for the non-skin cells, if stem cell functions are to be invoked, rather than mere, albeit curious and fascinating, co-integration events. This is especially important given that the cell grafting technique involves supplying cultured cells into a pouch of an otherwise complete skin (besides the epidermal/dermal split). Along the same lines, a more challenging grafting technique in form of the chamber graft with competent dermal cells and

without host skin epithelial cells, or even the patch assay of intracutaneous injection with competent dermal cells, could serve that purpose and reveal the true potential of the non-skin epithelial cells.

Regarding the transplants of small tissue biopsies from R26-LacZ reporter mice, two points are unclear. In the cornea and vagina samples the ORS is unlabeled, while inner layers appear LacZ+ from donor cells. It is unexplained how this is even possible when the entire tissue was transplanted. The foot pad makes sense and entirely labels apparently a full follicle. In either case it is unclear how only epithelial cells are labeled and mesenchymal cells are unlabeled even though they should be donor derived. Is the entire mesenchyme from the host?

2. The role of p63 is unconfirmed and does not make it here a p63 paper. It is highly doubtful that in all tissues all basal layer cells of slow and fast cycling stem cells and downstream transit amplifying cells are indeed all p63 positive. As such it cannot be excluded that p63 negative cells took hold in cultures that upregulated p63 in culture conditions. To claim that within any given tissue only p63 cells but not p63- epithelial cells have plasticity and can contribute to epidermis, seb gland or hair, both p63+ and - cells needs to be isolated, cultured and grafted separately, e.g. from p63 reporter mice or mice that were lineage marked by p63-Creer or the like. I am aware that GFP rats were used, but mouse epithelial cells should work as well. Conversely, misexpression of p63 in p63- cultured duodenal epithelial cells could be tested if it can confer plasticity. I am aware that this experimentation is likely beyond a reasonable timeline for this story, but in the absence of any functional p63 manipulations the p63 link is correlative at best. Without further experimentation, the authors should tone down the p63 angle a lot and make it a discussion point at the most.

Other points:

1. It would be very beneficial to trace back and confirm the expression of donor tissue related transcription factors in the epidermis and follicles, identified by microarray/RNAseq
2. Abstract: It seems an overstatement that "epithelial stem cells regardless of origin can response to skin morphogenetic signals and contribute to epidermis, sebaceous glands and hair follicles". Based on the data presented neither all cells contribute nor do they do so to all tissues.
3. Fig. 2d. the heat map code is not clear when it is labeled "pathways". What are different levels of pathways and their units?
4. To better demonstrate the incorporation of transplanted GFP+ cells in Fig. 1d into the basal layer of epidermis, and into hair follicle bulge, matrix and differentiated lineages, it would be important to include co-staining of GFP+ positive cells with corresponding well-known markers.
5. The interpretation of the PCA in Fig. 2A is not convincing. The data are meant to show that a switch in gene expression towards hair follicle fate (with exception of trachea-derived cells) occurs in cells recovered months after transplantation. How does it work that 2nd round of culture cells are closer to cells from whisker, even though they graft much less efficient? More importantly even 2nd round cultured trachea cells are close to whisker, even though they do not give rise to hair follicles. On a technical note, the presented PC1 and PC2 cover less than 50% of variance only.

Reviewer #1 (Remarks to the Author):

In this manuscript, Claudinot et al., studied the plasticity of p63-expressing epithelial cells derived from various tissues including all three germ lines (ectoderm, endoderm and mesoderm). The authors transplanted (with and without culturing) p63-expressing cells from various tissues onto back skin. The authors further claim that the transplanted cells survived and contributed to hair follicles and epidermis. Interestingly, the authors found that transplantation of p63-expressing cells from central cornea, limbus, oral mucosa, foot pad, vagina, esophagus, bladder and whisker bulge but not the cells isolated from trachea. The authors also claim that certain keratin genes that are characteristics to tissue of origin are gradually lost and they instead acquired the markers of host tissue. Furthermore, the authors performed RNA-seq analysis on cells isolated from first and second round of transplantation and directly compared them to that of cells before transplantation and found that transcriptomes of transplanted cells are more similar to that of skin hair follicles and the epidermis while losing the characteristics of tissue of origin. Interestingly, they found that p63-expressing cells derived from tissues such as trachea did not contribute to hair follicle regeneration. Instead, the transplanted cells formed contributed to stratified squamous epithelium of the interfollicular epidermis.

These findings are of significant interest in milieu of understanding the mechanisms of cellular plasticity and harnessing the potential of plasticity for cell-based therapies. However, to further strengthen these findings the authors should address the following concerns.

1. Based on the presence of labeled transplanted cells in hair follicles the authors state that cells isolated from various organs have the ability to generate hair follicles and interfollicular epidermis. However, the presence of labeled transplanted cells is a mere indication of cells filling the gaps in the damaged tissue. Therefore, the authors should check markers of various compartments of the hair follicles. For example, the authors should check the expression of SOX9, KRT15, LGR5, LHX2 and GLI1 to determine whether p63-expressing cells derived from other tissues generate/contribute to appropriate cell types in different compartments of the hair follicle?

We agree with the reviewer that it is an important point. It is our mistake not to have discussed it properly in this report because we thought that have already addressed it in previous publications (Claudinot et al., PNAS 2005; Bonfanti et al., Nature 2010). In these reports, we have unambiguously demonstrated using clonal analysis and serial transplantation that cultured epithelial cells isolated from the whisker of the rat (Claudinot et al., PNAS 2005) or from thymus (Bonfanti et al., Nature 2010) can contribute to the formation and renewal of sebaceous glands and pelage hair follicles for the entire life of the animal. The transplanted EGFP positive rat cells integrate in different proportions the developing mouse hair follicles. The rat cells can then generate all the differentiated lineages (8 different lineages) that are necessary to form a functional hair follicle. Most importantly, this process is maintained through numerous hair cycles and in serial transplantation (the most stringent stem cell assay). Clonal analysis and functional assays clearly demonstrate that the transplanted cells can behave as *bona fide* multipotent stem cells of the hair follicles in response to proper skin morphogenetic signals.

We have performed immunocytochemistry to investigate the expression of a number of genes mentioned by the reviewer including Sox9, Lhx2, Gli1 and Cux1. There is no reliable antibody for Lgr5 according to Nick Barker, our colleague in Singapore. Despite numerous attempts (different antibodies, different experimental conditions), we have been unable to obtain a reliable data, possibly because the antibodies did not properly recognize rat cells.

As stated above, we strongly believe that the functional demonstration that the EGFP positive rat cells can generate all lineages of the hair follicles after more than 200 days of transplantation (several hair cycles) fully answers the reviewer concern. To further illustrate this point, we have modified Figure 1 and integrated a panel (d) showing several beautiful hair follicles entirely EGFP positive generated from transplanted EGP dome cells from the rat bladder, 238 days after transplantation. This result can only be obtained if the EGFP rat cells behave as *bona fide* multipotent stem cells of the hair follicle. We have also

included a new panel (Panel C supplementary figure 3) showing similar results obtained with cells isolated from the trigone of the bladder and from the prostate.

2. In Figure 3b, p63-expressing cells derived from only pluristratified epithelia contribute to hair follicles but not from cells derived from pseudostratified, transitional or simple epithelium. The authors should check whether the markers of donor tissue continue to express when they do not generate/contribute to hair follicles. Such analysis would aid in better understanding why certain p63-expressing cells from some tissues do not exhibit plasticity.

We fully agree with the reviewer, but it is impossible to perform any immunohistochemistry to detect the markers of donor tissues because the beta-galactosidase activity was revealed by chemical reaction (Xgal staining) and sections were performed in historesin.

3. Along the similar lines, the authors should utilize their RNA-seq data to map the differences in p63-expressing cells derived from trachea vs other pluristratified organs. Although the functional validation of potential candidates is not in the scope of the current manuscript, mapping the differences in transcriptome will aid in understanding the mechanisms of plasticity in future studies.

We agree with the reviewer that it will be interesting to search for genes which expression is different in trachea but that it also out of the scope of this manuscript. This said, we have compared gene expression in cultured epithelial tracheal cells to other cultured TP63 expressing epithelial cells in our RNA-seq data using gorilla software. Before transplantation, only 66 genes were significantly up or down regulated. No specific process, function or component enrichment could be singled out. After transplantation, 57 genes were significantly up or down regulated, and again no specific process, function or component enrichment could be singled out.

4. In figure 5a, the authors state that developmental origin of vaginal squamous epithelium is mesoderm. However, it should be noted that distinct parts of the vaginal epithelium are derived from endoderm or mesoderm during development. Therefore, the authors should specifically state which part of the vaginal epithelium was used to isolate p63-expressing cells.

We have included the following sentence “Only the proximal vagina (internal third, excluding the cervix), the thoracic part of the oesophagus and the distal trachea were used” in the paragraph Tissue dissection and cell dissociation in Supplementary Methods

5. Plasticity associated with cell transplantation or cell culture has been studied in many tissues. More specifically, epithelial cells isolated from sweat glands have been shown to acquire the characteristics of mammary epithelial cells when the host animal undergoes microenvironmental changes due to pregnancy and lactation. Such plasticity was not observed in normal hosts. The current findings are similar to prior work (Lu et al., Cell, 2012). It is important to acknowledge these prior findings.

We have referenced this paper and added a sentence in the discussion: “or transplantation of sweat gland epithelial progenitor into hormonally-induced mammary gland microenvironment during pregnancy and lactation (Lu et al., 2012).

Reviewer #2 (Remarks to the Author):

The authors conduct a comprehensive analysis of the capacity of Tp63 expressing cells to produce hairy skin when placed in the appropriate niche. This is an interesting and important story. The volume of in vivo studies is impressive, however the analysis of the extensive but disparate global gene expression data requires significant revision in terms of presentation and downstream analysis that will improve

manuscript significantly.

Integrating data analysis from multiple array platforms and RNA-seq is challenging. Details of how these were compared are not given. Please clarify in both text and in detail in methods? Was analysis limited to comparison of samples on single platform and then comparison of differential lists?

We have clarified this point by adding several explanatory sentences in the text and in supplementary information

Text page 6:

Quantitative analysis of RNA expression was then performed using the Affymetrix GeneChip Rat Expression Array 230 2.0 (31,042 gene-level probe sets) or Rat Gene ST 1.0 Array (27,342 gene-level probe sets). The results of the thymus analysis that was used for comparison were published previously and deposited at the NCBI Gene Expression Omnibus public (<http://www.ncbi.nlm.nih.gov/geo>) database (series record GSE21686)³ (Supplementary fig. 6a to 6d). The global gene expression in the different tissues was compared to the whisker hair follicle and list of genes which are significantly up- or downregulated were established (fold-change >2; p-value < 0.05).

The global gene expression in the different tissues was compared to the whisker hair follicle and list of genes which are significantly up- or downregulated were established (fold-change >2; p-value < 0.05).

Supplementary information

Page 7

Statistical gene analysis was performed in R (version 3.3.2). Genes with low counts were filtered out with a minimal threshold of one count per million (cpm) in at least one sample. Library sizes were scaled using TMM normalization (EdgeR package, v. 3.14.0)⁵⁴ and log-transformed with Limma Voom (Limma package v. 3.28.21)⁵⁵. Differential expression, with whisker follicle as reference, [A1] was analyzed with Limma⁵⁶ and a moderated t-test was applied for each comparison. Genes were computed by the Benjamini-Hochberg method, controlling for false discovery rate. Genes are selected as differently expressed with a log₂ Fold-change higher or lower than 2 and -2 respectively, as well with an adjusted p-value <0.05. Lists of genes significantly up- or downregulated were constituted and compared with Excel (Microsoft), and pathways were analyzed by Gorilla software [A2] (www.cbl-gorilla.cs.technion.ac.il). For quantitative RT-PCR, one mg of RNA from FACS-sorted cultured cells was reverse transcribed using the superscript III reverse transcriptase kit (Invitrogen). Each sample of cDNA from the 1st and 2nd culture was amplified in Technical triplicates [A3] using the Taqman Universal Mastermix II, no UNG (Applied Biosystem) on a QuantStudio 6 Flex Real-Time PCR System (ThermoFisher). Data analysis was performed using the Expression Suite Software v.1.0.4 [A4] (Applied Biosystems) using TBP, SDHA and Tubb as normalizers. Relative quantity (RQ) expression data were extracted as a single value from the Expression Suite analysis and [A5] reported in GraphPad Prism7, using the whisker bulge cells expression from the 1st round of culture as reference. Primers (Applied Biosystems) are listed below.

In figure 2C what is the significance of overlaps?

We have modified the text to answer this question.

“Venn diagram showing the overlap of the down- and up-regulated genes common to vaginal, thymic, bladder (dome) and tracheal cells in the 1st and 2nd rounds of culture, fold-change >2, adjusted p-value <0.05. d) Heat map for the expression of some representative genes of different pathways important for hair follicle development

The transcriptome of cells cultured from various epithelia was compared to that of the whisker before and after transplantation into the skin microenvironment. The number of genes that are significantly up- or downregulated in each tissue vs the hair follicle is written in each circle, the overlaps indicating the

number of common genes between 2 comparisons (e.g: 139 genes are common between the comparisons Vagina vs HF and Bladder vs HF).

The authors mention their pathways analysis did not detect WNT pathways. What did this identify? The results are not illustrated or described? Did they identify p63 enrichment of p63 targets?

We have searched for genes that were up- and downregulated in cells other than hair follicle. Genes from the canonical Wnt pathway were especially investigated, as they are implicated in the hair follicle morphogenesis and hair cycling (Gat et al., 1998; DasGupta et Fuchs, 1999). No specific enrichment for the Wnt pathway was observed before or after the transplantation using the gorilla ontology analysis software (<http://cbl-gorilla.cs.technion.ac.il/>).

We agree with the reviewer that it will be very interesting to perform a thorough analysis of p63 targets. However, a thorough study will need more experiments, significant bioinformatics and collaboration with specialists in the field. We believe that is out of the scope of the paper.

Nevertheless, as preliminary results, we have searched for genes that were up- and downregulated in cells other than hair follicle ($p < 0,05$; $FC > 2$) in a list of Trp63-targets or Trp63-bound genes primary or cultured cells in human (Barton et al., 2010; Truong et al., 2006; Yang et al., 2006) or mouse (Ortt et al., 2008). 14 genes in a list of 34 mouse target genes (Ortt et al., 2008) showed up but only 3 of them (*Dnmt3a*, *Ivl*, *Synj2*) showed up in all comparisons. These genes were mainly linked to metabolism (arachidonic acid, glucathione, vitamin A), immune system or different processes in the cell. *Calml5*, *Il1a*, and *Phlda1* are common to all these studies on human keratinocytes. *Calml5* is implicated in multiple signaling (adrenergic, calcium or estrogen signaling) as well as in aldosterone synthesis. *Il1a* is involved in apoptosis signaling pathways, cytokine and inflammatory response, and hematopoietic cell lineage. *Phlda1* has a role in the anti-apoptotic effects of insulin-like growth factor-1 and cell-cycle.

We have also searched for genes reported as direct p63 targets and p63-bound genes in signaling pathways or enriched in adhesion categories (Yang et al., 2006). Twelve genes were identified but only 2 genes (*Areg* and *Dusp1*) were differentially expressed in all samples. These genes are implicated in cell proliferation and cell cycle.

Moreover they then show what I believe to be array expression of a number of individual genes in the heatmap in figure 2d but the scale is labelled “pathways”?

We have modified the legend for better comprehension: (b) Expression clustering analysis. (c) Venn diagram showing the overlap of the down- and up-regulated genes common to vaginal, thymic, bladder (dome) and tracheal cells in the 1st and 2nd rounds of culture, fold-change > 2 , adjusted p-value < 0.05 .

The authors should additionally evaluate the role of direct p63 target genes in regulating the differentially genes they identify either through overlapping with published p63 ChIP-seq data or through use of publicly available or commercial tools such as ENRICHR or IPA which provide quick ways to evaluate this.

As stated above, we fully agree with the reviewer that it will be very interesting to perform a thorough analysis of p63 targets. However, a thorough study will need more experiments, significant bioinformatics and collaboration with specialists in the field. We believe that is out of the scope of the paper.

The authors refer to “Memory” of transplanted cells on page 10 line 253. This is suggestive of an epigenetic program that is maintained. Have the authors looked at changes in DNA methylation in any of these analyses? Moreover, given queries raised above relating to p63 target genes, what are the transcription factors driving expression of these maintained genes and are these also enriched for p63 targets? i.e. are the Tp63 positive cells primed for expression for squamous cells as would be suggested from extensive analysis from CHIP-seq studies that suggests Tp63 bookmarks squamous specific enhancers.

It would be intriguing to map these transcriptional trajectories onto the schematic in Figure 5b and I think it may be possible with data contained herein.

We have not looked at DNA methylation. Again we agree with the reviewer that this will be important important experiments, and again we believe that they are out of the scope of this paper.

Figures 1C/E why are there no error bars? Are these single PCR reactions? It appears these results were not biologically replicated? But only technically replicated? Legends should state this and include standard deviation of technical replicates. Is this more compelling than merely relying on array data and either way does this then warrant inclusion of this many panels in main figures?

Detailed were added in the supplementary information (page 7). Fold-change was given by the software as a single value, excluding the presence of an error bar.

Data analysis was performed using the Expression Suite Software v.1.0.4 [A4] (Applied Biosystems) using TBP, SDHA and Tubb as normalizers. Relative quantity (RQ) expression data were extracted as a single value from the Expression Suite analysis and [A5] reported in GraphPad Prism7, using the whisker bulge cells expression from the 1st round of culture as reference.

Labels of these PCR panels are very hard to read and would benefit from a colour coding? This is also true for Figure 2e. perhaps the colour coding could be matched to 2a PCA plot. The same is true for figure 2b where same colour scheme should be used for labels.

We have also added colours as suggested (Fig 2 panel b). Adding colours to panel d makes the panel non readable

With regard to aneuploid cells and their capacity to transplant is this true when tissue are exposed to DNA damage. E.g. in UV containing skin wherein mutated clones containing for example p53 mutations may pre-exist is there a potential for transplantation to induce expansion of the potentially cancer pre-cursors?

Interesting question but a completely different project

What is the clone forming capacity in vitro (Holo/Meri/Parclones of tracheal cells compared to other cell types?

Cells were harvested from rat tissues, where this denomination cannot be used (Barrandon and Green PNAS 1987). Moreover, rodents cells have a tendency to immortalize when they are serially passaged (Reichert et Haase, 2010). The plating efficiency of trachea is relatively lower (1,75%, passages II to IV) than for other tissues (Vagina 3,375% to Prostate 22%).

Nomenclature for p63 should be consistent. Human p63 is TP63, Rat Tp63 and mouse is Trp63 this should be checked carefully throughout as well as same for other genes.

As we work with rat cells, the notation used in all the manuscript is Trp63, even when the rat cells are transplanted into a mouse skin microenvironment. We have modified the nomenclature in the following sentence as it referred to mouse Trp63

” As expected, Trp63-negative stem cells of the duodenum²² could not grow under a microenvironment designed for epidermis but could nicely expand to form mini-guts under appropriate culture conditions¹⁸”

The authors should change “genome-wide screen” to something that reflects this is analysis of gene expression e.g. “global transcriptional analysis”

Done

Reviewer #4 (Remarks to the Author):

In this manuscript, Claudinot et al., demonstrate contribution of epithelial cells derived from non-skin p63+ epithelia to epidermis and/or sebaceous glands and hair follicles. Furthermore, with microarray and RNA-seq analysis they show that although cells can integrate they remain a molecular memory in the form of donor tissue transcription factor expression. Also, that Wnt signaling factors are beneficially expressed, possibly facilitating, as they speculate, the conversion to hair follicle and epidermal cells. Finally, they show that the default integration capacity depends on epithelial classification, and that cell culture conditions can unmask the potential of non-squamous epithelial cells to integrate with epidermis and hair follicles. Overall, this is an interesting study with the curious observation that non-hairy epithelial cells have the ability to integrate into hair and/or epidermis. However, two major concerns regarding the considerably loose interpretation of potential p63 functions and stem cell behavior dampen my enthusiasm.

Major concerns:

1. A major concern is that the authors invoke stem cell functions and translational application prospects which is a big stretch, although they were otherwise careful with wording when they describe the grafting success as that cultured cells “contribute” or “nicely integrated”. It appears that percent of integrations refers success of integration at any detectable level, but does not reflect percent of full formation potential. As remarkable as “contribution” and “integration” of non-skin epithelial cells is, it does not at all prove competence. From the images provided in Fig. 1d for bladder and Fig S3b for vagina and prostate, for example, most parts of ORS and infundibulum appear unlabeled. For the bladder cells, the hair follicle structures can be barely recognized and some larger form of growing masses are seen instead. Unsurprisingly, the non-hairy, but footpad skin-derived cells performed much better. As long as it is not 100% of grafted cells within a given larger area of epidermis, or the entire epithelial complement of single hair follicles (bulge, matrix, all diff lineages), competence to induce and/or maintain the tissue is not established. As such stem cell functions implied for the grafted epithelial cells are not established. The authors have previously shown in Ref 32 remarkable competence of transplanted bulge to supply the entire follicle unit in successive cycles. Nothing less should be employed to show competence for the non-skin cells, if stem cell functions are to be invoked, rather than mere, albeit curious and fascinating, co-integration events. This is especially important given that the cell grafting technique involves supplying cultured cells into a pouch of an otherwise complete skin (besides the epidermal/dermal split). Along the same lines, a more challenging grafting technique in form of the chamber graft with competent dermal cells and without host skin epithelial cells, or even the patch assay of intracutaneous injection with competent dermal cells, could serve that purpose and reveal the true potential of the non-skin epithelial cells.

On Fig1D, we replaced the macroscopic view by a high magnification of the hair follicles formed by the transplanted cells. A panel was also added in supplemental figure 3. We can clearly see that “the entire epithelial complement of single hair follicles (bulge, matrix, all diff lineages)” is composed by transplanted cells, meaning that “competence to induce and/or maintain the tissue is established”, as the referee #4 asked. Moreover, the long-term transplantation (238 days for bladder cells) indicates that the transplanted cells behave as multipotent stem cells: they contributed to the renewal and the differentiation in hair follicles and sebaceous glands, as well as epidermis, for the entire life of the animal.

As asked by the reviewer, the technique labelled as reference 32 (Oshima et al, 2001), was indeed used in figure 3 to demonstrate the potentialities of the various epithelial tissues (see question 2 below). In both techniques we used (Claudinot et al., 2005 and Oshima et al., 2001), the cultured cells and the full-thickness tissues were respectively put in contact of inductive dermis and epidermis from the neonate mouse skin.

The technique we used to transplant cultured cells was previously published (Claudinot et al., 2005) and is the only one (as far as I know) which can be used for such a long period of time.

Regarding the transplants of small tissue biopsies from R26-LacZ reporter mice, two points are unclear. In the cornea and vagina samples the ORS is unlabeled, while inner layers appear LacZ+ from donor cells. It is unexplained how this is even possible when the entire tissue was transplanted. The foot pad makes sense and entirely labels apparently a full follicle. In either case it is unclear how only epithelial cells are labeled and mesenchymal cells are unlabeled even though they should be donor derived. Is the entire mesenchyme from the host?

As mentioned above, the technique referenced as 32 (Oshima et al., 2001) was used for this figure. Full-thickness pieces of LacZ adult tissues, containing epithelium and mesenchyme, were inserted into the skin of neonate mice. But we never observed any zone of host mesenchyme with blue labelling, meaning that either the donor mesenchyme was eliminated or the bgal gene switched off (unknown promoter, Jax datas). As in the fig 4D of ref 32, the proportion of labelled cells in the hair follicles can be variable, in the different lineages of the hair follicles. One of the most probable hypotheses is that cells are able to migrate from the donor tissues to the recipient skin and contribute in different proportion to the hair follicle morphogenesis.

2. The role of p63 is unconfirmed and does not make it here a p63 paper. It is highly doubtful that in all tissues all basal layer cells of slow and fast cycling stem cells and downstream transit amplifying cells are indeed all p63 positive. As such it cannot be excluded that p63 negative cells took hold in cultures that upregulated p63 in culture conditions. To claim that within any given tissue only p63 cells but not p63-epithelial cells have plasticity and can contribute to epidermis, seb gland or hair, both p63+ and - cells needs to be isolated, cultured and grafted separately, e.g. from p63 reporter mice or mice that were lineage marked by p63-Creer or the like. I am aware that GFP rats were used, but mouse epithelial cells should work as well. Conversely, misexpression of p63 in p63- cultured duodenal epithelial cells could be tested if it can confer plasticity. I am aware that this experimentation is likely beyond a reasonable timeline for this story, but in the absence of any functional p63 manipulations the p63 link is correlative at best. Without further experimentation, the authors should tone down the p63 angle a lot and make it a discussion point at the most.

This paper is about functionality of TP63 expressing cells. The role of TP63, as correctly stated by the reviewer is not addressed in this paper (see answer to reviewer 2). The fact that Tp63 null mice die at birth precludes a number of classical experiments to address the functionality of Tp63 expressing cells. This question could be possibly addressed in the future using gene editing technology on cultured cells but it is out of the scope of this paper.

Our results show that Tp63-expressing epithelia contains cells which are able to respond to the neonate skin microenvironment, what Tp63 negative tissues (gut, uterus), are not. In the Tp63-positive tissues, immunohistochemistry shows that almost all the cells from the basal layer, express the protein, even with a high-diluted antibody. Moreover, only Tp63-expressing tissues contain clonogenic keratinocytes in our culture system, making impossible to test the Tp63-negative cell potential in this technique. *In vitro*, all the cells of the colonies express the Tp63 protein, except the differentiated cells from the centre of the colony, which stratify. In the first days of culture, no Tp63-negative cells were observed (double staining Hoechst/Tp63, Chromatin staining really easy to distinguish between the rat and mouse feeder cells).

Other points:

2. Abstract: It seems an overstatement that “epithelial stem cells regardless of origin can respond to skin morphogenetic signals and contribute to epidermis, sebaceous glands and hair follicles”. Based on the data presented neither all cells contribute nor do they do so to all tissues.

3. Fig. 2d. the heat map code is not clear when it is labeled “pathways”. What are different levels of pathways and their units?

Answered in reviewer 2

4. To better demonstrate the incorporation of transplanted GFP+ cells in Fig. 1d into the basal layer of epidermis, and into hair follicle bulge, matrix and differentiated lineages, it would be important to include co-staining of GFP+ positive cells with corresponding well-known markers.

Answered in reviewer 1

5. The interpretation of the PCA in Fig. 2A is not convincing. The data are meant to show that a switch in gene expression towards hair follicle fate (with exception of trachea-derived cells) occurs in cells recovered months after transplantation. How does it work that 2nd round of culture cells are closer to cells from whisker, even though they graft much less efficient? More importantly even 2nd round cultured trachea cells are close to whisker, even though they do not give rise to hair follicles. On a technical note, the presented PC1 and PC2 cover less than 50% of variance only.

The efficiency of transplantation has no impact on the PCA as cells from the 2nd round of culture are originated from the transplantation. In this technique, entire transplant was dissociated, but only GFP rat keratinocytes can be serially cultured, as the conditions are not suitable for mouse cells. Moreover, for RNAseq, as well as for microarray, EGFP epithelial cells were FACS sorted.

Transcriptome analysis shows that there is a relative and incomplete shift towards the hair follicle program. As mentioned in fig 4a, some critical transcription factors, specific of the donor tissue, were still expressed after transplantation, showing a “memory” of the origin. The cells after transplantation seem to be in an “intermediate” state between their origin and the hair follicle.

Reviewers' comments:

Reviewer #1 (Remarks to the Author):

I think the authors have addressed all the comments and provided experimental data or explanation. The author have made attempts to experimentally address all comments. However, in response to on my comment (#1), the response was rather disappointing. One of the key point in this manuscript is that Tp63+ cells from various organs can acquire skin epithelial cell characteristics when transplanted on the skin. The authors failed to provide molecular evidence that transplanted cells acquire epidermal molecular programs. The authors state that antibodies for some of the key epidermal markers did not work on cells from rat origin. However, there are many previous reports which showed staining for these markers.

In my view, it is very important to know whether TP63+ cells from other organs can fully convert into skin epidermal cells. My suggestion is that the authors could use single molecule FISH to detect transcripts if the antibodies are not working on rat-derived cells.

Reviewer #2 (Remarks to the Author):

In their responses to my comments, Claudinot et al. have addressed some of the issues I raised. In addition to ongoing issues not addressed in response to reviewers 1 and 3 who are clearly better placed to comment on the in vivo studies. My major outstanding issue concerns clarity of presentation of valuable microarray and RNA-seq datasets and the availability of the normalised gene expression matrices and genelists to enable readers to reproduce and expand on authors analysis. I have focussed my comments on things that are definitely within this scope and achievable and if these remaining concerns are addressed adequately along with any outstanding issues raised by the authors reviewers 1 and 3 are addressed then a revised manuscript may be suitable for publication in Nature Comms.

1) While some additional information regarding the primary analysis of the two datasets is provided. It would also be useful to explain how they (tried) to made the array and RNA-seq datasets compatible for comparison in figure 2.

2) Given responses to some of later comments to additional data analysis, it is crucial that the lists of significantly altered genes that underpin analyses in Figure 2 and summary normalised matrices for microarray and RNA-seq need to be included in supplementary material to enable readers to examine gene-list and look at other genes of interest in these valuable datasets.

3) I also noted in the legend for Figure 2 it only mentions RNA-seq not microarray?

4) I also asked "In figure 2C what is the significance of overlaps?" the authors have clarified the significance of the list not the overlaps? I would like to know what the likelihood of the overlaps between lists are. This can be computed by Fishers or hypergeometric test as described here <https://rdr.io/bioc/GeneOverlap/man/GeneOverlap.html>. Importantly these tests need to consider the overlap between the RNA-seq and Array platforms in terms of number of genes that could have been detected

5) In general the responses to questions regarding further analysis using public data are disappointing,

however I understand that describing what they may have done or could do could dilute message of paper, hence if all genelists are made available to allow readers to carry out their own analysis
6) the authors have not answered my question regarding technical and biological replicates in RT-PCR data

I asked "Figures 1C/E why are there no error bars? Are these single PCR reactions? It appears these results were not biologically replicated? But only technically replicated? Legends should state this and include standard deviation of technical replicates. Is this more compelling than merely relying on array data and either way does this then warrant inclusion of this many panels in main figures?"

They responded "Detailed were added in the supplementary information (page 7). Fold-change was given by the software as a single value, excluding the presence of an error bar."

Do bars represent the mean of three technical replicates? Or are they the mean of the means of 3 samples from each tissue type? If the former, it is a concern that these conclusions may not be reliable. If the latter it is certainly remiss to not include error bars. Either way it needs to be stated. Moreover if results are replicated in either way it is possible to include error bars of SD in former and SEM in latter to show variability of data.

Reviewer #4 (Remarks to the Author):

In this revision, Claudinot et al were able to address most of my concerns. For example, the new magnified images in Fig. 1D from a long-term transplantation is able to demonstrate the ability of transplanted cells to form the entire hair follicle. However, there are still a few concerns that are needed to be addressed:

1. In Fig. 1D, the magnified views do convincingly show the contribution of transplanted cells to the entire hair follicle. However, the contribution of the transplanted cells in 2nd transplantation to the hair follicles or within a given larger area of epidermis is remarkably weak in Fig. 1D. This is surprising given the effective culture expansion of GFP isolated cells from the 1st transplants. How would the authors not expect better performance when the non-skin cells were able to form hair stem cells in vivo in the 1st transplant and supposedly had further matured in this way towards hair? In this case they should perform better one would expect. Therefore, the claim of successful serial transplantation of the non-skin epithelial cells may be a stretch. The first transplant, highly efficient and newly presented, with robust long-term engraftment is impressive in itself though.

2. The provided explanation for the lack of LacZ in mesenchyme of whole grafted reporter tissue pieces are very weak. The ubiquitous expression of R26 promoter is well known and the migration of 100% mesenchyme is unlikely. It would mean that the entire mesenchyme from grafted skin is replaced by the host. This is a big weakness and unknown in an otherwise well-put together important paper.

3. The new QPCR data for lineage transcription factors demonstrating hair fate are appreciated and make sense. However, IF staining for a few of them in GFP+ grafts would be much more appropriate to directly link grafted GFP+ cells to the markers, on a cell by cell basis. Location location location.

Reviewer #1 (Remarks to the Author):

I think the authors have addressed all the comments and provided experimental data or explanation. The author made attempts to experimentally address all comments. However, in response to my comment (#1), the response was rather disappointing. One of the key points in this manuscript is that TP63+ cells from various organs can acquire skin epithelial cell characteristics when transplanted on the skin. The authors failed to provide molecular evidence that transplanted cells acquire epidermal molecular programs. The authors state that antibodies for some of the key epidermal markers did not work on cells from rat origin. However, there are many previous reports which showed staining for these markers.

In my view, it is very important to know whether TP63+ cells from other organs can fully convert into skin epidermal cells. My suggestion is that the authors could use single molecule FISH to detect transcripts if the antibodies are not working on rat-derived cells.

We have addressed this question by performing immunocytochemistry experiments using a new set of antibodies. We have demonstrated that TP63 positive/EGFP positive epithelial cells cultured from the bladder and the prostate of the rat expressed proteins linked to hair follicle stem cells and hair differentiation (Sox9, Lhx2, Krt15 and Krt-31) at the proper location (figure 2 and supplementary figure 3). These results together with the fact that the EGFP rat-derived hair follicles are structurally normal with all the hair layers present, properly organized and functional (producing a hair) (figure 2), demonstrate that the transplanted cells of non-hairy origin can fully embark in a hair follicle program in response to a hairy skin microenvironment.

Reviewer #2 (Remarks to the Author):

In their responses to my comments, Claudinot et al. have addressed some of the issues I raised. In addition to ongoing issues not addressed in response to reviewers 1 and 3 who are clearly better placed to comment on the in vivo studies. My major outstanding issue concerns clarity of presentation of valuable microarray and RNA-seq datasets and the availability of the normalised gene expression matrices and gene lists to enable readers to reproduce and expand on authors analysis. I have focussed my comments on things that are definitely within this scope and achievable and if these remaining concerns are addressed adequately along with any outstanding issues raised by the authors reviewers 1 and 3 are addressed then a revised manuscript may be suitable for publication in Nature Comms.

1) While some additional information regarding the primary analysis of the two datasets is provided. It would also be useful to explain how they (tried) to make the array and RNA-seq datasets compatible for comparison in figure 2.

In the figure 2, only results of the RNAseq analysis are represented, as mentioned in the legend. These results are not compared with the microarray data, which are presented in supplemental figure 6. According to our biostatisticians, it is impossible to compare results of the microarray and the RNA-Seq because the two techniques were performed on different samples, with different tissues, at different timings and with different techniques. We have uploaded all the microarrays and RNA-Seq data for readers' appreciation (see below).

2) Given responses to some of later comments to additional data analysis, it is crucial that the lists of significantly altered genes that underpin analyses in Figure 2 and summary normalised matrices for microarray and RNA-seq need to be included in supplementary material to enable readers to examine gene list and look at other genes of interest in these valuable datasets.

The full dataset of the significant genes in the RNA-Seq and microarray analysis was added to the manuscript and can be found in supplementary tables 3 and 5. The details of the 2 experiments are published at the NCBI Gene Expression Omnibus public database (#GSE21686, # GSE116717 and #GSE116719).

3) I also noted in the legend for Figure 2 it only mentions RNA-seq not microarray?

Figure 2 only presents the results of the RNA-Seq analysis.

4) I also asked "In figure 2C what is the significance of overlaps?" the authors have clarified the significance of the list not the overlaps? I would like to know what the likelihood of the overlaps between lists are. This can be computed by Fishers or hypergeometric test as described here <https://rdrr.io/bioc/GeneOverlap/man/GeneOverlap.html>. Importantly these tests need to consider the overlap between the RNA-seq and Array platforms in terms of number of genes that could have been detected

As mentioned above, figure 2 is only about RNA-Seq data; there is no comparison between RNA-Seq and microarray data. The Venn diagrams show the overlap of the down- and up-regulated genes common to vaginal, thymic, bladder and tracheal cells in the 1st and 2nd rounds of culture. An analysis using the Gorilla software did not point to an enrichment of specific pathways or function.

5) In general the responses to questions regarding further analysis using public data are disappointing, however I understand that describing what they may have done or could do could dilute message of paper, hence if all gene lists are made available to allow readers to carry out their own analysis

Please refer to answer n°2

6) the authors have not answered my question regarding technical and biological replicates in RT-PCR data I asked "Figures 1C/E why are there no error bars? Are these single PCR reactions? It appears these results were not biologically replicated? But only technically replicated? Legends should state this and include standard deviation of technical replicates. Is this more compelling than merely relying on array data and either way does this then warrant inclusion of this many panels in main figures?"

Figures were modified and ΔCt with SEM were presented, rather fold-Change. Details are provided in the legends. As mentioned in the supplemental material and methods, each biological sample was run in technical replicates for the qPCR reaction.

They responded "Detailed were added in the supplementary information (page 7). Fold-change was given by the software as a single value, excluding the presence of an error bar." Do bars represent the mean of three technical replicates? Or are they the mean of the means of 3 samples from each tissue type? If the former, it is a concern that these conclusions may not be reliable. If the latter it is certainly remiss to not include error bars. Either way it needs to be stated. Moreover if results are replicated in either way it is possible to include error bars of SD in former and SEM in latter to show variability of data.

Error bars represent the mean of three technical replicates. We have added this information in the figure legends. For this experiment, GFP rat cells from different non-hairy tissues were cultivated in triplicate onto a feeder layer of lethally irradiated mouse 3T3-J2 cells and the GFP cells were then sorted by FACS to eliminate the residual mouse cells. To make the cell sorting doable, the FACS facility decided for each tissue to pool the triplicate cultures together (16 different cultures and a 3T3 control). RNAs were then extracted and used for RNA-Seq and RT-qPCR. The number of samples that were then processed for RT-qPCR was overwhelming (17 samples x 3 technical replicates x 24 probes = 1224 samples).

Reviewer #3 (Remarks to the Author):

In this revision, Claudinot et al were able to address most of my concerns. For example, the new magnified images in Fig. 1D from a long-term transplantation is able to demonstrate the ability of transplanted cells to form the entire hair follicle. However, there are still a few concerns that are needed to be addressed:

1. In Fig. 1D, the magnified views do convincingly show the contribution of transplanted cells to the entire hair follicle. However, the contribution of the transplanted cells in 2nd transplantation to the hair follicles or

within a given larger area of epidermis is remarkably weak in Fig. 1D. This is surprising given the effective culture expansion of GFP isolated cells from the 1st transplants. How would the authors not expect better performance when the non-skin cells were able to form hair stem cells in vivo in the 1st transplant and supposedly had further matured in this way towards hair? In this case they should perform better one would expect. Therefore, the claim of successful serial transplantation of the non-skin epithelial cells may be a stretch. The first transplant, highly efficient and newly presented, with robust long-term engraftment is impressive in itself though.

Many factors can impact the efficiency of engraftment. As an example, the number of serial passages necessary to expand the initial cell population is a crucial point as the multiplication of serial passages can lead to a spontaneous immortalization of rodent cells (Reichelt and Haase, 2010). Hence, we usually did not culture the cells for more than five passages. Immortalization has a direct impact on engraftment as we have showed in this manuscript. Moreover, some tissues are more prone to immortalization for unknown reasons, and the proportion of polyploid cells can also vary from culture to culture, again for unknown reasons. We have also found that there is no strict correlation between the number of clonogenic cells in of a culture and the efficiency of engraftment. As the reviewer rightly pointed it out, one would expect a higher efficiency in the second round of transplantation; we do not have a good explanation for that observation. A thorough study of the factors impacting engraftment would certainly be interesting, but it is a several years research program, out of the scope of this paper.

2. The provided explanation for the lack of LacZ in mesenchyme of whole grafted reporter tissue pieces are very weak. The ubiquitous expression of R26 promoter is well known and the migration of 100% mesenchyme is unlikely. It would mean that the entire mesenchyme from grafted skin is replaced by the host. This is a big weakness and unknown in an otherwise well-put together important paper.

We have never observed LacZ-positive mesenchymal cells that remain after transplantation in more than 200 experiments. One explanation is that these cells are diluted out after transplantation and thus hard to observed. Another explanation is that LacZ mesenchymal cells of donor origin are selected against. We have never observed dermal papilla that were from the donor (LacZ positive); we have included a sentence in the text indicating that the dermal papilla was always from the host. We have also included a star in figure 4 (panel B central cornea) to show a lacZ negative dermal papilla. We fully agree with the reviewer that a thorough study of the behavior of mesenchymal cells after transplantation would be interesting as little is known about it. Again, it is a several years research program, out of the scope of this paper.

3. The new QPCR data for lineage transcription factors demonstrating hair fate are appreciated and make sense. However, IF staining for a few of them in GFP+ grafts would be much more appropriate to directly link grafted GFP+ cells to the markers, on a cell by cell basis.

Please refer to the answer to Reviewer 1.

REVIEWERS' COMMENTS:

Reviewer #1 (Remarks to the Author):

The authors have addressed all comments.

Reviewer #2 (Remarks to the Author):

The revisions the authors have made based on my suggestions and those of other reviewers have greatly improved the clarity of the manuscript. The changes to presentation and description of the RNA-seq data and qRT-PCR addresses most of my concerns and I have some minor outstanding issues, which if addressed

1) I would like to review a .txt file or excel version of supplementary table-3 to ensure this is useful for readers.

2) I would also ask for access code/token to review GEO submissions for GSE116717 and #GSE116719.

3) The authors have still not fully understood my question (4) "I also asked "In figure 2C (now 3C) what is the significance of overlaps?" the authors have clarified the significance of the list not the overlaps? I would like to know what the likelihood of the overlaps between lists are. This can be computed by Fishers or hypergeometric test as described here <https://rdr.io/bioc/GeneOverlap/man/GeneOverlap.html>.

The authors have clarified that this is all RNA-seq and suggest "An analysis using the Gorilla software did not point to an enrichment of specific pathways or function.". This is not what I was asking, rather simply are the number of overlapping genes between significantly up- or down-regulated lists in each comparison more than you would expect by random chance.

Reviewer #4 (Remarks to the Author):

I thank the authors for the thoughtful answers to my concerns. The authors addressed all of my points

Answers to reviewer 2

Reviewer #2 (Remarks to the Author):

The revisions the authors have made based on my suggestions and those of other reviewers have greatly improved the clarity of the manuscript. The changes to presentation and description of the RNA-seq data and qRT-PCR addresses most of my concerns and I have some minor outstanding issues, which if addressed

1) I would like to review a .txt file or excel version of supplementary table-3 to ensure this is useful for readers.

Supplementary table 3 in xlsx is attached

2) I would also ask for access code/token to review GEO submissions for GSE116717 and #GSE116719.

The following secure token has been created to allow review of record GSE116717 while it remains in private status: krabumeepnkzncz

The following secure token has been created to allow review of record GSE116719 while it remains in private status: yhsvqewqrpkvjwx

3) The authors have still not fully understood my question (4) “I also asked “In figure 2C (now 3C) what is the significance of overlaps?” the authors have clarified the significance of the list not the overlaps? I would like to know what the likelihood of the overlaps between lists are. This can be computed by Fishers or hypergeometric test as described here <https://rdr.io/bioc/GeneOverlap/man/GeneOverlap.html>. The authors have clarified that this is all RNA-seq and suggest “An analysis using the Gorilla software did not point to an enrichment of specific pathways or function.”. This is not what I was asking, rather simply are the number of overlapping genes between significantly up- or down-regulated lists in each comparison more than you would expect by random chance.

The significance of overlap between lists of differentially expressed genes (adjusted p-value ≤ 0.05 , absolute fold change ≥ 2) before transplantation and following transplantation was calculated using the hypergeometric distribution (phyper function in R). Briefly, lists were compared pairwise by inputting the number of intersecting genes (i), number of genes in comparison 1 (m), the number of genes in comparison 2 (k) and the total number of expressed genes (N) into the phyper function in R and p-values calculated for depletion and enrichment. The results showed that overlaps between the lists both before and after transplantation are significantly enriched (the probability that the number of genes in the overlap was at least as much as that observed was zero or close to zero in all cases). The results of the hypergeometric test are provided in Excel tables ‘Enrichment_Before_Transplantation_withKey.xlsx’ and ‘Enrichment_After_Transplantation_withKey.xlsx’.

We shall include this text in the final revised version of the paper as well as the attached files.